# Prevalence of Anxiety, Depression, and Insomnia Among Medical Workers in Emergency Medical Services in Eastern Kazakhstan

**DOI:** 10.3390/ijerph22030407

**Published:** 2025-03-10

**Authors:** Diana K. Kussainova, Ainash S. Orazalina, Zaituna A. Khismetova, Dinara Serikova-Esengeldina, Zaituna G. Khamidullina, Kamila M. Akhmetova, Anar E. Tursynbekova, Assel R. Tukinova, Gulnar M. Shalgumbayeva

**Affiliations:** 1Department of Public Health, Semey Medical University, 103 Abay St., Semey 071400, Kazakhstan; diana.kussainova@smu.edu.kz (D.K.K.); ainash.orazalina@smu.edu.kz (A.S.O.); zaituna.khismietova@mail.ru (Z.A.K.); dinara.esengeldina@smu.edu.kz (D.S.-E.); tukinova_asel@mail.ru (A.R.T.); 2Department of Public Health, Astana Medical University, Beybitshilik Street 49a, Astana 010000, Kazakhstan; zaituna59@gmail.com (Z.G.K.); kamila_maratovna@list.ru (K.M.A.); 3Department of Quality Control of Medical Services, City Clinical Hospital No. 5, Dostyk Avenue, 220b, Almaty 010017, Kazakhstan; dr.tursynbekova@gmail.com

**Keywords:** sleep disorders, mental health, work-related stress, emergency medical station personnel, Central Asia, Eastern Europe

## Abstract

**Introduction**: Studying the prevalence of anxiety, depression, and insomnia among medical workers in emergency medical services is a relevant task that will improve our understanding of scope of the problem and develop effective strategies to support and prevent psychological problems among medical staff. Insomnia is closely linked to anxiety and depression, as sleep disturbances can exacerbate emotional distress, while persistent anxiety and depressive symptoms contribute to sleep disruptions. Individuals suffering from insomnia are at a higher risk of developing anxiety and depression, creating a bidirectional relationship that negatively impacts overall mental well-being. This raises a crucial question: “What specific measures and intervention strategies can be implemented to reduce the levels of anxiety, depression, and insomnia among EMS personnel?” **Methods**: A cross-sectional study was conducted with the participation of 592 medical workers employed in emergency medical services in the East Kazakhstan and Abay regions of the Republic of Kazakhstan. This study included questions regarding the socio-demographic data of the respondents, questions assessing the severity of insomnia using the Insomnia Severity Index (ISI), and questions from the Hospital Anxiety and Depression Scale (HADS) scale assessing the level of anxiety and depression among the participants. **Results**: Nearly a third of the EMS personnel reported symptoms of insomnia (28.2% subthreshold, 16.2% insomnia, and 3.0% severe), anxiety (22.1% subclinical, and 13.0% clinical), or depression (20.4% subclinical, and 9.8% clinical). Feldshers (nursing staff) and those with higher education had elevated levels of these conditions. The insomnia was strongly correlated with anxiety (r = 0.539, *p* < 0.001) and depression (r = 0.415, *p* < 0.001), emphasizing the need for targeted mental health interventions. **Conclusions**: This study found elevated levels of insomnia, anxiety, and depression among emergency medical service (EMS) personnel—especially nursing staff and those with higher education. We recommend comprehensive mental health support, routine screenings, stress management training, and integrating sleep hygiene into wellness programs.

## 1. Introduction

Healthcare workers play a key role in providing patient care, but they are also subject to high levels of stress and psychological strain that can lead to the development of anxiety, depression, and insomnia [1]. Emergency Medical Services (EMS) play a crucial role in delivering immediate, life-saving care during emergencies, and their unique working conditions expose them to higher levels of stress and psychological strain [2].

Depression is a serious psychological disorder characterized by a constant feeling of sadness, loss of interest in life, fatigue, sleep and appetite disturbances, and a negative impact on a person’s daily life. Prolonged depression can lead to decreased immunity, poor physical health, and an increased risk of cardiovascular disease [3].

Studies have shown that depression is highly prevalent among healthcare professionals. For instance, a systematic review reported a global prevalence ranging from 22.8% to 38.9%, with frontline medical staff being particularly vulnerable [4]. In China, 50.4% of healthcare workers reported symptoms of depression during the COVID-19 pandemic [5], while in Europe the rates among physicians and nurses vary between 19% and 34%, depending on the region and specialty [6].

Similarly, EMS personnel are at high risk [7]. In the United States, studies have found depression rates of 6.8% and 15% among EMS personnel [8], and in Brazil, 32.6% of EMS personnel experienced moderate to severe depression [9].

Anxiety is defined as a state of restlessness, tension, and nervousness that can range from mild anxiety to panic attacks [10]. It may be accompanied by physiological symptoms such as rapid heartbeat, sweating, and trembling, while constant stress and anxious thoughts can elevate cortisol levels and negatively affect the cardiovascular, digestive, and immune systems [11].

Insomnia is a sleep disorder where a person experiences difficulty falling asleep, interrupted sleep, or premature awakening. It can be caused by stress, anxiety, depression, or other factors and significantly affects cognitive function, concentration, and memory. Moreover, sleep disturbances increase the risk of both depression and anxiety [12].

Biologically, insomnia disrupts the circadian rhythm, leading to an imbalance in stress hormones like cortisol, which in turn is associated with heightened anxiety and depression [13].

The study of anxiety, depression, and insomnia among EMS personnel is highly relevant. EMS personnel face high levels of stress and traumatic situations, encountering emergencies, unexpected deaths, and severe injuries [12]. Psychological problems in healthcare providers not only affect their well-being but also impair the quality of care by increasing the risk of medical errors and treatment failures [14]. Addressing these issues can help develop supportive measures and training programs to strengthen the psychological resilience and emotional well-being of medical personnel [15].

EMS personnel often face long shifts, night duties, and irregular schedules, leading to disrupted sleep patterns that cause insomnia, which in turn contributes to anxiety and depression [16]. The disruption of the circadian rhythm from irregular sleep patterns results in hormonal imbalances (e.g., elevated cortisol), further exacerbating psychological distress. International studies revealed that long working hours and excessive workload directly contribute to poor sleep quality and increased stress and depression [17].

Over the past two decades, the Emergency Nurse Practitioner (ENP) role has become an integral part of emergency care. It was developed to address growing healthcare demands, mitigate ongoing medical workforce shortages, and meet key emergency care delivery targets [18].

The study of Norway showed that recent health policy changes have significantly altered the role of nurse leaders, especially in rural settings where complex emergency services are provided. It highlights that nurse leaders now face additional responsibilities for implementing and innovating services, which require them to integrate clinical expertise with organizational and managerial skills [19]. Portuguese scientists, in the study “Construction of a Professional Competency Matrix of the Nurse in Emergency Services”, argue that integrating clinical and organizational skills is essential for achieving excellence in emergency care [20]. The study of innovation technology in the Emergency Nurse Service found that insufficient organizational support, excessive restrictions, and underuse compromise service sustainability, underscoring that robust integration is essential for future success [21].

In Kazakhstan, significant changes since 2017 in the emergency medical service system have increased working hours for many EMS staff, including feldshers and paramedics [22]. The proportion of feldsher (nursing staff) teams has notably increased, with the ratio of physician-led teams to feldshers at 18% to 82% as of 2021 [23].

In Kazakhstan, recent legal reforms—outlined in the law “On the Approval of Rules for the Provision of Emergency Medical Care, Including the Use of Medical Aviation” [24]—have standardized EMS operations. Under this framework, EMS dispatchers must triage calls via the “103” hotline within five minutes according to urgency categories. This regulatory change has increased workload and stress levels among EMS personnel, contributing to psychological strain such as depression and anxiety.

Thus, studying the prevalence of anxiety, depression, and insomnia among EMS personnel is a relevant task that will allow us to better understand the scope of the problem and develop effective strategies to support and prevent psychological problems among medical staff. To further ground our work in the existing literature, we now outline the key theoretical concepts that underpin our study, namely the biopsychosocial model of health, which highlights the interplay between biological, psychological, and social factors in the development of mental health issues; stress and coping theories, particularly the transactional model of stress, to explain how EMS personnel manage work-related stress; and the circadian rhythm theory, which underpins our understanding of sleep disorders and their impact on mental health. This investigation aimed to study the prevalence of anxiety, depression, and insomnia among EMS personnel in the eastern region of Kazakhstan. We hypothesize that EMS personnel in Eastern Kazakhstan exhibit a high prevalence of anxiety, depression, and insomnia. Additionally, we expect a significant positive correlation between insomnia severity and the levels of anxiety and depression.

## 2. Materials and Methods

### 2.1. Design

The East Kazakhstan region occupies the easternmost part of Kazakhstan, and as of 2020, has a population of 1,369,597 [25]. The Abay region was previously known as the Semipalatinsk region. Upon Kazakhstan’s independence in 1991, the Semipalatinsk region continued to exist until 1997, when it was merged back into the East Kazakhstan region. In 2022, the region was reconstituted as the Abay region, with its administrative center located in Semey (Semipalatinsk). The total population of the region is 638,300 [25].

This was a cross-sectional, questionnaire study carried out in the East Kazakhstan and Abay regions, between September and December 2022.

### 2.2. Participants and Settings

The Emergency Medical Stations in the East Kazakhstan and Abay regions are public healthcare institutions that provide emergency medical services. The study sample was drawn from these public emergency services, which are responsible for delivering care to both adult and pediatric populations in life-threatening conditions, accidents, and severe acute illnesses. These institutions are part of the state healthcare system and not the private sector.

In Kazakhstan, the EMS system is designed to provide rapid and effective pre-hospital care, ensuring that patients receive timely medical assistance in critical situations. The system comprises several key occupational roles, each with specific training and responsibilities. Physicians working in EMS in Kazakhstan are highly trained medical doctors who have completed a full medical degree followed by specialized training or residencies in emergency medicine, trauma care, or critical care. Their responsibilities include advanced patient assessment, performing emergency procedures, and making critical clinical decisions in pre-hospital settings. Their expertise is crucial in managing complex medical emergencies. Feldshers serve as mid-level healthcare providers in Kazakhstan’s EMS, particularly in regions where the immediate presence of a physician may not be feasible. Their training is typically vocational or technical in nature, focusing on practical skills such as basic diagnostics, first aid, patient stabilization, and the administration of essential medications. Feldshers are responsible for providing prompt, efficient care in emergency situations, thereby supporting the overall EMS team by ensuring that initial treatment is delivered quickly and effectively. Paramedics in Kazakhstan receive extensive training that covers advanced life support techniques, trauma management, cardiac care, and emergency patient stabilization. Training programs for paramedics are designed to equip them with the skills needed to rapidly assess patients, perform life-saving interventions, and safely transport patients to hospitals. Emergency vehicle drivers can serve as paramedics if they have received the appropriate training and possess the necessary first aid skills. In addition to the clinical staff, the EMS system in Kazakhstan includes dispatchers who are trained to triage emergency calls and coordinate rapid responses. They play a critical role in managing communication between the emergency site and the EMS teams, ensuring that resources are allocated efficiently. Together, these roles form a comprehensive emergency services framework in Kazakhstan, designed to address a wide range of medical emergencies through a coordinated and well-trained team approach.

This study employed a convenience sampling approach due to practical limitations, including participant accessibility and the specific focus on emergency medical workers in the East Kazakhstan and Abay regions. This method allowed for an efficient data collection process within the targeted population. We acknowledge its limitations in generalizing findings to the broader EMS workforce; however, the insights gained remain valuable for understanding the mental health challenges faced by emergency medical workers in these regions.

The study sample was drawn from EMS personnel in the East Kazakhstan and Abay regions. The total number of participants in the study was 592. In the East Kazakhstan Emergency Medical Station work 438 EMS personnel, of whom 320 gave consent for the study. In the Abay Emergency Medical Station work 493 EMS personnel, of whom 272 provided consent.

Participants were approached directly on site during their scheduled shifts by members of our research team, who explained the study details and obtained informed consent. Participation was entirely voluntary and was not a compulsory part of their work. The inclusion criteria encompassed EMS personnel who were adult responders (aged ≥18 years), healthcare professionals (physicians, feldshers (nurses), and paramedics), and able to complete the study questionnaire. Participants were excluded if they refused to participate, or were EMS personnel in other regions of Kazakhstan.

Data were gathered via a self-administered paper questionnaire, which was distributed by the researchers at each station. All participants in this study gave written consent after receiving detailed information about its purpose and the confidentiality of their personal data. Each participant was assigned a unique code, and a file linking this code to their personal identification information was kept by the database custodian, who was the only one with access to it. Others had access only to the coded (secure) database. This study had prior approval from the Ethics Committee of Semey Medical University (Protocol No. 4, 20 December 2021).

### 2.3. The Questionnaires

This study included questions regarding the socio-demographic data of the respondents, and questions assessing the severity of insomnia using the Insomnia Severity Index (ISI) [26]. The ISI is a 7-item self-report questionnaire designed to evaluate the nature, severity, and impact of insomnia. Each item is rated on a 5-point Likert scale ranging from 0 (no problem) to 4 (very severe problem), resulting in a total score between 0 and 28. The score interpretation is as follows: 0–7 indicates no insomnia, 8–14 suggests sub-threshold insomnia, 15–21 reflects moderate insomnia, and 22–28 signifies severe insomnia.

In addition, the study included questions from the Hospital Anxiety and Depression Scale (HADS) to assess the level of anxiety and depression among participants. The HADS comprises 14 items, divided into two subscales: anxiety and depression. Each item is scored on a four-point scale, with the maximum possible score being 21 for each subscale. A score of 11 or higher on either subscale indicates a significant case of psychological morbidity, while scores between 8 and 10 are considered borderline, and scores from 0 to 7 are classified as normal. In this study, the HADS was utilized to measure the emotional well-being of emergency medical workers in the Eastern Kazakhstan region [27].

The Hospital Anxiety and Depression Scale (HADS) consists of two subscales: HAD-A (Anxiety) and HAD-D (Depression). Each subscale includes 7 items, specifically designed to measure anxiety and depression symptoms separately. The HAD-A subscale assesses symptoms related to anxiety, such as restlessness, fear, and tension, while the HAD-D subscale focuses on symptoms of depression, including feelings of sadness, loss of interest, and hopelessness.

Each item on both subscales is rated on a four-point scale, ranging from 0 to 3, where the scores are as follows: 0 = Not at all; 1 = From time to time, occasionally; 2 = Frequently, in the last week; 3 = Most of the time, or nearly always. The scores for each subscale are summed, with a maximum possible score of 21 for both the HAD-A and HAD-D subscales. Based on the total score, the following classifications are used: 0–7: Normal (no significant symptoms of anxiety or depression); 8–10: Borderline (subclinical symptoms); 11 or higher: Clinically significant symptoms (indicating a possible case of anxiety or depression).

Each participant provided responses to every question, with each response being assigned a numerical score based on a predetermined scoring system. The scores for all items were then summed to obtain a total score for each participant. This total score was subsequently used to assess and interpret the severity of symptoms according to established criteria.

The “forward/backward” procedure was applied to translate the ISI and HADS questionnaires from English into the Kazakh and Russian languages. Four general practitioners translated the questionnaires into Kazakh and Russian and these were backward translated into English by a health professional and a professional translator. Then, provisional versions of the Kazakh and Russian questionnaires were provided. All authors developed the final version.

The reliability was assessed by measuring the internal consistency of the questionnaires using Cronbach’s alpha coefficient, with a value of 0.70 or higher deemed satisfactory. For the Hospital Anxiety and Depression Scale (HADS), the Cronbach’s alpha for the anxiety subscale was 0.74 and for the depression subscale was 0.66. Similarly, for the Insomnia Severity Index (ISI), the Cronbach’s alpha was 0.70. These values indicate that both instruments demonstrate good internal consistency within our study sample. The final drafts of both the Kazakh and Russian versions were administered to a sample of the study population.

### 2.4. Data Analysis

Descriptive analysis was conducted and the continuous data was summarized as means and standard deviation (SD), and the categorical data as frequencies and proportions. To justify the use of parametric tests, we conducted normality tests on the data. Specifically, we applied the Shapiro–Wilk test and Kolmogorov–Smirnov test to assess whether the data followed a normal distribution. The results of both tests indicated that the data conformed to the assumptions of normality (*p* > 0.05). Additionally, we visually inspected the data using Q-Q plots, which further confirmed that the data points closely followed the expected line, supporting the conclusion of normal distribution. Groups were compared using an independent *t*-test, and one-way analysis of variance (ANOVA). Bonferroni correction was applied to post hoc tests to adjust for multiple comparisons. Pearson’s correlation analysis was used to examine the relationships between insomnia severity (ISI) and anxiety and depression scores (HADS). Correlation coefficients (r) and *p*-values were calculated, with significance set at *p* < 0.05. SPSS version 20.0 program (IBM Ireland Product Distribution Limited, Ireland) was used for statistical analysis.

## 3. Results

### 3.1. Sample Description

The study included 592 medical staff who worked in the emergency medical stations of the study regions. The mean age of the respondents was 37.0 (±12.5). The characteristics of the participants at baseline are shown in Table 1.

The group of respondents aged between 26 and 35 years comprised 33.3% (*n* = 197), while the group older than 46 years was 29.6% (*n* = 175). The majority of the respondents were male with 70.4% (*n* = 417). The survey was conducted among EMS personnel living in two regions: East Kazakhstan—54.1% (*n* = 320) and Abay—45.9% (*n* = 272). The majority of respondents had secondary education—83.1% (*n* = 492) and worked as feldshers (nursing staff)—82.1% (*n* = 486) (Table 1).

### 3.2. Prevalence of Insomnia, Anxiety, and Depression

Table 2 presents the interpretation of the results of insomnia, anxiety, and depression levels.

The study found that 28.2% (*n* = 167) of the respondents had subthreshold insomnia, 16.2% (*n* = 96) had insomnia, and 3.0% (*n* = 18) had severe insomnia. It was found that 22.1% (*n* = 131) of the health care workers had subclinically expressed anxiety, and 13.0% (*n* = 77) showed clinically expressed anxiety. Subclinically expressed depression was revealed at 20.4% (*n* = 121) of EMS personnel and the remaining 9.8% (*n* = 58) showed clinically expressed depression. The majority of participants in our study fall into the non-clinically significant subgroups across all measures (ISI, HADS-A, HADS-D). Specifically, 52.5% of participants in the ISI group, 64.9% in the anxiety subscale (HADS-A), and 69.8% in the depression subscale (HADS-D) were classified as non-clinically significant.

### 3.3. Comparison by Participant Characteristics

In Table 3, we present the results of the ANOVA and *t*-test to compare the differences in outcome measures (ISI, HAD-A, HAD-D) based on participant characteristics. In the following Table 4, we provide the results of post hoc pairwise comparisons within the groups, with Bonferroni correction applied to adjust for multiple comparisons.

The comparison of the participants’ outcome measures based on the participant characteristics revealed significant differences between groups with accordance to specialty and level of education. The post hoc tests with Bonferroni correction confirmed statistically significant pairwise differences (Table 4). Feldshers (nursing staff) had a higher level of insomnia at 12.0 (±7.5), anxiety at 7.4 (±4.4), and depression at 6.7 (±4.2). However, participants with a higher level of education predominantly showed elevated levels of insomnia at 11.2 (±7.3) compared with participants with secondary level education at 7.6 (±6.6). The EMS personnel with higher levels of education also had elevated levels of anxiety at 7.2 (±4.3) and depression at 6.3 (±4.1) compared with colleagues with secondary education.

A comparison between groups was conducted using post hoc tests with Bonferroni correction. In this post hoc analysis, multiple pairwise comparisons were conducted across different age groups and professional subgroups. No statistically significant differences were found between any age groups in terms of ISI, HADS-A, or HADS-D scores (all *p*-values > 0.05). Statistically significant within-group differences were identified for professional subgroups (Table 4). This table presents the post hoc comparisons of the ISI, HADS-A, and HADS-D between three groups: physicians, feldshers (nursing staff), and paramedics (drivers). Statistically significant results (*p* < 0.05) are highlighted below, considering the Bonferroni correction. Physicians demonstrated significantly lower ISI scores compared with feldshers (nursing staff), with a mean difference of −4.42 (95% CI: −6.30, −2.54; *p* < 0.001). Feldshers (nursing staff) reported significantly higher ISI scores than paramedics, with a mean difference of 5.03 (95% CI: 1.13, 8.94; *p* = 0.006). Physicians reported significantly lower anxiety scores compared to feldshers (nursing staff), with a mean difference of −1.72 (95% CI: −2.81, −0.62; *p* = 0.001). Physicians demonstrated significantly lower depression scores than feldshers (nursing staff), with a mean difference of −1.34 (95% CI: −2.41, −0.35; *p* = 0.004).

### 3.4. Correlation Analysis

Pearson’s correlation was used to examine the relationship between the severity of insomnia and anxiety and insomnia and depression in the sample of 592 healthcare workers in EMS.

Table 5 shows the correlation matrix for the variables, with the Pearson correlation coefficients and corresponding *p*-values. The correlation was statistically significant at the 0.01 level (two-tailed).

A significant positive correlation was found between insomnia severity and anxiety (r = 0.539; *p* < 0.001). This suggests that higher levels of insomnia are strongly associated with increased anxiety levels. A significant positive correlation was observed between insomnia severity and depression (r = 0.415; *p* < 0.001), indicating that higher levels of insomnia are associated with higher levels of depression. A significant positive correlation was observed between anxiety and depression (r = 0.561; *p* < 0.001), indicating that higher levels of anxiety are associated with higher levels of depression

## 4. Discussion

### 4.1. Main Findings and Interpretation

Our study reveals that insomnia is highly prevalent among EMS personnel with a significant portion of participants exhibiting clinically significant symptoms. Specifically, 28.2% of EMS personnel had subthreshold insomnia, while 16.2% had clinically significant insomnia and 3.0% were severely affected. Importantly, 52.0% did not experience any significant sleep disruptions. These findings reflect varying levels of insomnia severity with higher rates likely attributable to factors such as long and irregular work hours, high stress levels, and emotional and physical demands of emergency work. Additionally, our data corroborate previous research demonstrating that insomnia contributes to poor mental health outcomes like anxiety (r = 0.539) and depression (r = 0.415). Our results show a positive correlation between insomnia severity and both anxiety and depression.

Insomnia is a common sleep problem that can affect healthcare workers as well [28]. Insomnia can be particularly problematic due to their irregular work schedules, stress, high levels of responsibility, and the need to remain alert and focused during long shifts [29]. Insomnia in healthcare workers can lead to poorer quality of work, fatigue, irritability, memory and attention problems, and an increased risk of errors and accidents [30]. Insomnia can exacerbate anxiety and depression through various mechanisms, such as the hyperactivation of the hypothalamic–pituitary–adrenal (HPA) axis, which leads to increased cortisol production, neurochemical imbalances, including alterations in serotonin and gamma-aminobutyric acid (GABA) levels, and disturbances in circadian rhythms, which further affect mood regulation and stress responses [31]. Therefore, healthcare workers must focus to their sleep and take care of their physical and emotional well-being.

Healthcare professionals can employ various strategies to combat insomnia such as maintaining a regular sleep schedule, creating a calm and comfortable sleep environment, avoiding caffeine and other stimulants before bedtime, engaging in relaxation practices such as yoga or meditation, and seeking professional help if insomnia becomes a chronic problem [28]. According to our findings, a significant proportion of EMS personnel exhibited symptoms of insomnia. Specifically, 28.2% of participants had subthreshold insomnia, indicating mild symptoms that do not yet meet the criteria for a clinical diagnosis. Additionally, 16.2% of the participants experienced insomnia with clinically significant symptoms, while 3.0% were severely affected by insomnia, indicating a high level of disruption to their sleep and overall functioning. However, it is important to note that 52% of participants did not report significant sleep disturbances. These findings reflect the varying levels of insomnia severity among the study population.

The higher rates of insomnia could be attributed to various factors inherent to the nature of the profession, such as irregular working hours, high stress levels, and frequent exposure to emergency situations. Studies have shown that EMS personnel are particularly vulnerable to sleep disorders due to their demanding schedules and the emotional and physical toll of their work [32].

Additionally, 52% of the participants did not report significant sleep disturbances. This could be explained by the varying resilience of individuals to work-related stressors and their ability to maintain healthy sleep habits despite the pressures of their occupation. Factors such as social support, coping mechanisms, and lifestyle choices (e.g., physical activity, relaxation techniques) could contribute to better sleep quality among these individuals [33].

Anxiety in EMS personnel is a common condition caused by stress, overwork, constant contact with patients and difficult cases, and the uncertainty and unknowns of the job. EMS personnel may experience anxiety due to high expectations, lack of resources, difficult decisions, and responsibility for patients’ lives. Constant stress and anxiety can lead to fatigue, emotional burnout, decreased productivity, and psychological problems. Therefore, EMS personnel must pay attention to their mental health, and seek support from colleagues, management, or mental health professionals. It is also important to practice relaxation techniques, exercise, and maintain a work–life balance to reduce anxiety and improve overall well-being [34].

### 4.2. Comparison with International Studies

Previous research indicates a higher prevalence of severe insomnia, anxiety, and depression in certain populations, such as EMS personnel, in other regions or countries [5,9,12]. The relatively low prevalence of these conditions in the EMS personnel of East Kazakhstan may be attributed to differences in healthcare management, levels of government support, working conditions, and workload. Further exploration of these factors is necessary to understand the unique context of this region and how it may influence mental health outcomes.

Various strategies can help combat insomnia, including maintaining a regular sleep schedule, creating a comfortable sleep environment, avoiding stimulants before bedtime, engaging in relaxation techniques such as yoga or meditation, and seeking professional help if insomnia becomes chronic [28].

Our findings indicate that the majority of EMS personnel (64.9%) demonstrated no clinically significant symptoms of anxiety, indicating a relatively low level of anxiety among most participants. However, a substantial proportion of participants exhibited varying degrees of anxiety. Specifically, 22.1% showed subclinical anxiety, meaning they had mild symptoms that did not reach the threshold for clinical diagnosis but were still noticeable. Additionally, 13.0% of participants experienced clinically significant anxiety, indicating that their symptoms were severe enough to potentially affect their daily functioning and require clinical attention.

Depression among EMS personnel is a serious problem that is often underestimated and under-discussed [35]. EMS personnel face high levels of stress, emotional and physical strain, constant pressure, and responsibility for the lives and health of their patients. Because of the constant contact with patients, difficult cases, and often traumatic situations, EMS personnel are at risk for developing depression. They may experience symptoms such as fatigue, despair, feelings of helplessness, loss of interest in work and life, and social isolation. Depression in healthcare workers should not be ignored. Help should be sought from psychologists, psychiatrists, or other professionals to diagnose and treat depression. It is also important to create a supportive environment in healthcare facilities where workers can communicate about their feelings and receive support and help. Combating depression in health care workers requires a comprehensive approach, including psychological support, training in stress management strategies, regular mental health monitoring, and creating conditions for a healthy work–life balance [36].

In our study, the majority of EMS personnel (69.8%) showed no significant symptoms of depression, indicating that most participants were not experiencing major depressive symptoms. However, we also observed that 20.4% of the participants exhibited subclinical levels of depression, meaning they reported mild symptoms that did not reach the threshold for a formal diagnosis but were still noteworthy. In contrast, 9.8% of participants experienced clinically significant depression, suggesting a more serious impact on their well-being and daily life, potentially requiring treatment or intervention.

### 4.3. Explanation of Possible Causes

Some of the socio-demographic factors associated with depression, anxiety, and insomnia in EMS personnel include high workload, lack of peer and management support, exposure to traumatic events, work–life imbalance, and occupational burnout. These factors can have a negative impact on the mental health of health care workers and increase the risk of developing psychological problems [37]. Investigating the prevalence of anxiety, depression, and insomnia among EMS personnel is important for several reasons. It is essential that caring for the mental well-being of EMS personnel be the center of attention, as they face high levels of stress and emotional strain in their work daily. Studying the psychological aspects of their condition will help to better understand their needs and develop effective support strategies [38]. The mental health of EMS personnel has a direct impact on their performance and ability to provide quality care [39]. Medical errors and deficiencies in the quality of care can result from mental health problems such as anxiety, depression, and insomnia. Studying these problems helps to identify factors that may negatively affect health care providers and offer appropriate solutions and support measures [40]. Mental health problems in EMS personnel can have economic consequences for the health care system. They can lead to decreased productivity, increased absenteeism from work, and increased treatment and rehabilitation costs. Studying the prevalence and factors associated with anxiety, depression, and insomnia provides an opportunity to assess the economic burden and develop interventions to reduce this burden [41]. Furthermore, studying the prevalence of anxiety, depression, and insomnia among EMS personnel helps to develop and tailor effective support and intervention programs. This may include stress management training, psychological support, creating a caring environment for mental health, and implementing measures to reduce the risk of developing mental health problems [42,43].

A study by Argentinian researchers showed a prevalence of anxiety at 44% and depression at 21.9% [44]. A study conducted by researchers from Saudi Arabia found that 32.3% of health care workers had high levels of anxiety and 36.1% had moderate levels [45]. A study conducted by Turkish researchers found that 77.6% of health care workers had depression, 60.2% had anxiety, and 50.4% had insomnia [46].

Our findings highlight notable differences in the outcome measures based on participants’ characteristics, specifically their specialty and education level. These differences suggest that both the nature of the healthcare role and the level of education may play a significant role in the severity of insomnia, anxiety, and depression among medical workers.

First, feldshers (nursing staff) were found to experience significantly higher levels of insomnia (12.0 ± 7.5), anxiety (7.4 ± 4.4), and depression (6.7 ± 4.2) compared with physicians and paramedics. These results could reflect the unique demands of the nursing staff role, which may involve more direct patient care and potentially higher emotional and physical stress. Meanwhile, paramedics (drivers) exhibited comparatively lower levels of insomnia, anxiety, and depression, which might be related to the specific nature of their duties, such as driving and transport, which may carry different stressors than direct medical care.

Additionally, our study revealed that EMS personnel with higher levels of education tended to report elevated scores for insomnia (11.2 ± 7.3), anxiety (7.2 ± 4.3), and depression (6.3 ± 4.1) compared to those with secondary education. This could suggest that while higher education provides medical knowledge and training, it may also lead to increased awareness of the psychological demands of EMS personnel, or could indicate greater self-reflection on mental health issues. Higher education may also correlate with roles that carry greater responsibility, which could lead to higher stress levels and more significant psychological strain.

These findings were further confirmed via post hoc tests with Bonferroni correction, which demonstrated significant differences in the severity of insomnia, anxiety, and depression between the groups. Physicians showed significantly lower levels of insomnia, anxiety, and depression than feldshers, while feldshers had higher scores than paramedics. These findings underscore the potential impact of the nature of medical specialties on mental health outcomes.

Our results are consistent with previous research indicating that EMS personnel, particularly those in direct patient care roles such as nursing staff, are at greater risk of experiencing psychological distress. The study revealed that critical incidents, along with workplace culture and demands, profoundly affect the mental, physical, and social well-being of ambulance personnel [47]. A Swiss investigation of emergency personnel and rescue workers found an increased prevalence of posttraumatic stress symptoms (PTSS). The study revealed that PTSS was strongly associated with psychological strain and increased suicidal ideation, reflecting the significant burden and diminished quality of life in this population [48]. A Norwegian investigation found that posttraumatic stress disorder (PTSD) is more prevalent among first responders—including Emergency Medical Services, fire service, and police force—than in the general population [49]. The higher prevalence of insomnia, anxiety, and depression among nursing staff and those with higher education may reflect both the emotional and cognitive demands of their roles, suggesting a need for targeted mental health support and interventions for these groups.

In this study, Pearson’s correlation analysis was used to explore the relationship between insomnia severity and both anxiety and depression in a sample of 592 EMS personnel. The results revealed a significant positive correlation between insomnia severity and anxiety (r = 0.539; *p* < 0.001), suggesting that higher levels of insomnia are strongly associated with increased anxiety levels. This finding supports the idea that insomnia and anxiety are often linked, with one potentially exacerbating the other. The strong correlation emphasizes the importance of addressing both conditions simultaneously in healthcare settings, where both are prevalent among medical workers.

Similarly, a significant positive correlation was observed between insomnia severity and depression (r = 0.415; *p* < 0.001). This indicates that higher levels of insomnia are associated with higher levels of depression. The relationship between insomnia and depression is well-established in previous research, as sleep disturbances can both result from and contribute to depressive symptoms. These findings suggest that addressing insomnia could be an effective strategy in mitigating depressive symptoms, particularly in populations experiencing high levels of psychological stress, such as EMS personnel.

These results are consistent with previous studies indicating strong links between sleep disturbances and mental health issues such as anxiety and depression. Given the significant associations found in this study, future research should investigate targeted interventions that focus on improving sleep quality as a means of reducing anxiety and depression in healthcare workers.

Studying depression, anxiety, and insomnia is important to the health care system for several reasons. First, these psychological problems can significantly impair a person’s physical health, leading to serious consequences such as an increased risk of cardiovascular disease [50]. Second, depression, anxiety, and insomnia can significantly impair a person’s quality of life, leading to social isolation, impaired occupational performance, and other negative consequences [51]. Finally, these psychological problems can cause people to seek medical care, which increases the burden on the health care system. Therefore, the study and effective management of depression, anxiety, and insomnia are important for maintaining public health and optimizing the health care system. A comprehensive approach is needed to solve the problem of anxiety, depression, and insomnia among health care providers [12]. Incorporating support programs such as counseling and group support can provide confidential and professional help [52]. Improving the work environment, such as reducing work hours and increasing staff numbers, can reduce stress levels and improve overall well-being [53]. Encouraging healthy lifestyles including physical activity, healthy eating, and regular sleep can help improve mental health [54]. Increasing awareness of mental illnesses and their signs and symptoms can help health care providers recognize and treat these conditions early. Solving the problem of anxiety, depression, and insomnia among healthcare workers will improve their mental health, increase productivity, and provide better patient care.

### 4.4. Strengths and Limitations of the Study

Our study had some limitations. Firstly, as this was a cross-sectional study, the causality between compared variables could not be established. Secondly, the surveyed population in this study was relatively small. Moreover, unknown and unmeasured confounders may exist, and the results should be interpreted with caution. Despite these limitations, it is the first study in Kazakhstan that has analyzed anxiety, depression, and insomnia among medical workers of emergency medical aid stations using valid instruments such as the ISI and HADS scale.

### 4.5. Recommendations for Practice and Research

Based on our findings, we recommend that EMS organizations optimize work schedules, provide stress management training and accessible mental health support, and establish standardized protocols for diagnosing and managing sleep disorders and psychological distress. Future research should use longitudinal designs to evaluate these interventions and explore additional factors, such as social support and individual coping strategies.

## 5. Conclusions

Our study revealed a low prevalence of insomnia, anxiety, and depression among EMS personnel, with significant differences based on specialty and education level. Feldshers (nursing staff) showed higher levels of these symptoms compared to physicians and paramedics, likely due to the emotional and physical demands of their roles. Additionally, EMS personnel with higher education reported more severe symptoms, potentially linked to increased responsibilities or self-awareness of mental health.

A significant positive correlation was found between insomnia severity and both anxiety and depression, suggesting that sleep disturbances may contribute to mental health issues. These findings highlight the need for targeted mental health interventions, particularly for nursing staff and those with higher education. Improving sleep quality may be an effective strategy for reducing anxiety and depression in EMS personnel.

Our study provides a critical foundation for future longitudinal research to further evaluate intervention effectiveness and address the underlying factors contributing to psychological distress in EMS personnel.

## Figures and Tables

**Table 1 ijerph-22-00407-t001:** Socio-demographic data of respondents.

Main Categories	Variable	Frequencies	Percent
Age group (years)	18–25	128	21.6
	26–35	197	33.3
	36–45	92	15.5
	46+	175	29.6
Gender	Male	417	70.4
	Female	175	29.6
Specialty	Physician	85	14.4
	Feldsher (nursing staff)	486	82.1
	Paramedic (driver)	21	3.5
Education	Higher	100	16.9
	Secondary	492	83.1
Region of residence	East Kazakhstan region	320	54.1
	Abay region	272	45.9

**Table 2 ijerph-22-00407-t002:** ISI and HADS scores: depression, anxiety, and sleep quality status among healthcare staff.

Interpretation	Frequencies	Percent
ISI		
No clinically significant insomnia	311	52.5
Subthreshold insomnia	167	28.2
Insomnia	96	16.2
Severe insomnia	18	3.0
HAD-A		
Absence of reliably expressed symptoms of anxiety	384	64.9
Subclinically expressed anxiety	131	22.1
Clinically expressed anxiety	77	13.0
HAD-D		
Absence of reliably expressed symptoms of depression	413	69.8
Subclinically expressed depression	121	20.4
Clinically expressed depression	58	9.8

ISI, Insomnia Severity Index; HADS-A, Hospital Anxiety and Depression Scale—Anxiety Subscale; HADS-D, Hospital Anxiety and Depression Scale—Depression Subscale.

**Table 3 ijerph-22-00407-t003:** Differences in outcome measures based on participant characteristics (*n* = 592).

Main Categories	Variable	Frequency (%)	ISI		HAD-A		HAD-D	
			Mean (SD)	*p* Value	Mean (SD)	*p* Value	Mean (SD)	*p* Value
Age group (years)	18–25	128 (21.6)	7.9 (6.7)	0.346	6.4 (3.7)	0.055	6.0 (3.5)	0.324
	26–35	197 (33.3)	8.5 (6.9)		6.4 (3.9)		6.0 (3.7)	
	36–45	92 (15.5)	8.4 (7.6)		5.6 (4.1)		5.0 (3.9)	
	46+	175 (29.6)	8,1 (6.6)		5.4 (3.9)		5.1 (3.5)	
Gender	Male	417 (70.4)	8.3 (6.7)	0.597	6.1 (3.9)	0.353	5.5 (3.7)	0.874
	Female	175 (29.6)	8.0 (7.1)		5.7 (4.0)		5.6 (3.5)	
Specialty	Physician	85 (14.4)	7.6 (6.5)	0.000	5.7 (3.8)	0.001	5.3 (3.5)	0.003
	Feldsher (nursing staff)	486 (82.1)	12.0 (7.5)		7.4 (4.4)		6.7 (4.2)	
	Paramedic (driver)	21 (3.5)	7.0 (6.5)		5.8 (3.3)		6.6 (3.6)	
Education	Higher	100 (16.9)	11.2 (7.3)	0.000	7.2 (4.3)	0.000	6.3 (4.1)	0.021
	Secondary	492 (83.1)	7.6 (6.6)		5.7 (3.8)		5.4 (3.6)	

ISI, Insomnia Severity Index; HADS-A, Hospital Anxiety and Depression Scale—Anxiety Subscale; HADS-D, Hospital Anxiety and Depression Scale—Depression Subscale; SD, Standard Deviation.

**Table 4 ijerph-22-00407-t004:** Bonferroni-corrected pairwise comparisons.

Variables	SI Mean Difference (95% CI)	*p*-Value	HAD-A Mean Difference (95% CI)	*p*-Value	HAD-D Mean Difference (95% CI)	*p*-Value
18–25 vs. 26–35	−0.05 (−0.31; 0.20)	1.000	0.01 (−1.17; 1.18)	1.000	−0.004 (−1.10; 1.09)	1.000
18–25 vs. 36–45	−0.07 (−0.38; 0.24)	1.000	0.76 (−0.65; 2.18)	0.922	0.91 (−0.41; 2.23)	0.419
18–25 vs. 46+	−0.04 (−0.30; 0.22)	1.000	0.95 (−0.2; 2.15)	0.224	0.89 (−0.23; 2.02)	0.217
26–35 vs. 36–45	−0.02 (−0.30; 0.27)	1.000	0.76 (−0.55; 2.07)	0.741	0.91 (−0.31; 2.13)	0.294
26–35 vs. 46+	0.01 (−0.22; 0.25)	1.000	0.95 (−0.13; 2.02)	0.120	0.90 (−0.11; 1.90)	0.111
36–45 vs. 46+	0.03 (−0.26; 0.32)	1.000	0.18 (−1.15; 1.52)	1.000	−0.01 (−1.26; 1.23)	1.000
Physician vs. Feldsher (nursing staff)	−4.42 (−6.30; −2.54)	0.000	−1.72 (−2.81, −0.62)	0.001	−1.34 (−0.41, −0.35)	0.004
Physician vs. Paramedic (driver)	0.62 (−2.95; 4.18)	1.000	−0.07 (−2.15; 2.00)	1.000	−1.26 (−3.20; 0.69)	0.366
Feldsher (nursing staff) vs. Paramedic (driver)	5.03 (1.13; 8.94)	0.006	1.64 (−0.63; 3.91)	0.252	0.12 (−2.01; 2.25)	1.000

ISI, Insomnia Severity Index; HADS-A, Hospital Anxiety and Depression Scale—Anxiety Subscale; HADS-D, Hospital Anxiety and Depression Scale—Depression Subscale; CI, Confidence interval.

**Table 5 ijerph-22-00407-t005:** Correlation matrix of insomnia severity and anxiety, insomnia severity and depression scores, and anxiety and depression.

Variables	Correlation Coefficient (r)	*p*-Value
ISI vs. HAD-A	0.539	0.000
SI vs. HAD-D	0.415	0.000
HAD-A vs. HAD-D	0.561	0.000

Correlation strength: 0.1–0.3: weak correlation; 0.3–0.5: moderate correlation; 0.5–1.0: strong correlation.

## Data Availability

The data presented in this study are available upon request from the corresponding author. The data are not publicly available due to privacy restrictions.

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
