# Peer review of "Prevalence of Anxiety, Depression, and Insomnia Among Medical Workers in Emergency Medical Services in Eastern Kazakhstan"

_ijerph, 2025, doi:10.3390/ijerph22030407_

Round 1
Reviewer 1 Report
Comments and Suggestions for Authors
Abstract
- Wrong spelling of ‘introduction’
- A short background on the relationship of insomnia and depression and anxiety should be mentioned.
- Spell out full term of ‘HADS’
Introduction
- Elaborate on the prevalence of depression among healthcare workers from previous studies or other regions, using several up-to-date references.
- Paragraph 4 – 7: Explanation of ‘Regulations on Emergency Medical Services’ and the rest of the information is not necessary to put in the introduction section.
- Instead, highlight the core problem on how the working hours affect the sleep quality of emergency health workers. Relate how insomnia and how it relatess to depression and anxiety from biological perspective.
- Highlight the research gap on population. Why is it important to study this issue among Eastern Kazakhstan population.
Methods
- “The Emergency Medical Stations of the East Kazakhstan and the Abay regions are independent healthcare institutions providing emergency medical services to both adult and pediatric populations in life-threatening conditions, accidents, and severe acute illnesses. “ – do authors intend to explain that the sample was obtained from private medical sectors? Please clarify
- “In the Emergency Medi-cal Station of the East Kazakhstan region, 438 healthcare workers are employed, of whom 320 (73.0%) confirmed their participation in the research.” – this statement is not necessary as the focus of this study is among emergency healthcare workers only.
- Please explain the sampling procedure in detail. From the article, it seems that it was done using non-probability sampling method. Please justify why authors opt for this instead of probability sampling.
- It is better to standardize the sample background as much as possible the variables, for example: only select those who had undergone certain hours of work, or those who have work in hospital for >5 years etc.
- “at each station.” – please elaborate further about the stations.
- Please include the ethic approval code obtained for this study.
- Please elaborate the first time use of the abbreviation “Hospital Anxiety and Depression Scale (HADS).”
- Please introduce the components of HADS as HAD-A and HAD-D and elaborate on the method of rating separately.
- Please elaborate the first time use of the abbreviation “standard deviation (SD)”
- Please include normality test analysis to justify the use of parametric tests mentioned.
Results
- Please rearrange the paragraphs properly according to the journal format.
- Results from Table 2 - The purpose of descriptive analysis is to tabulate and see the majority of the predominant subgroup. The authors should also mention the subgroup of “non-clinically significant” as they are the majority for all ISI and HADS.
- The way authors present Table 3 is not accurate as those data were analyze via ANOVA and post hoc test. A table listing pairwise comparisons between age groups, along with mean difference, confidence intervals, and adjusted p-values (after Bonferroni correction) should be created instead.
- Table 3 results – gender and education levels only have 2 subgroups. They are not supposed to be analyzed by ANOVA.
- The results of Table 4 is not necessary as the comparison has been made in Table 3.
- Please re-construct Table 5 and Table 6 by doing ISI vs HAD-A, ISI vs HAD-D etc. the table the should have separate column for correlation coefficient and significant value.
- If possible, please compare or correlate HAD-A and HAD-D as well.
- The authors should also highlight the strength of the correlation coefficient. 0.5 and 0.4 is considered average/moderate correlation, although they are statistically significant.
Discussion
- Please insert the references for the first few lines of the discussion.
- 2% is not nearly half, this statement is not accurate.
- “This indicating a notable prevalence of sleep disturbances among study population” – this statement is not accurate as based on the findings, only 3% were severely affected by insomnia while 52% were not affected.
- The authors only use one reference for the second paragraph while it is important to cite several up-to-date references when discussing about prevalence.
- The discussion are not done in-depth. There is lack of discussion to answer “WHY” parts of the results.
- 1) The readers would first like to understand how insomnia relates/lead to anxiety and depression pathologically. 2) the prevalence of severe insomnia, anxiety and depression are low in this population as compared to previous studies that were done using different population. Is there differences in healthcare management, government support, workload etc. 3) It is important to highlight and discuss the moderate strength of the correlation results instead of just the significant value.
Conclusion
- “Our study revealed a high prevalence of insomnia, anxiety, and depression among emergency medical service workers “ – this statement is not accurate based on the findings.
Author Response
Manuscript Revision
Journal: International Journal of Environmental Research and Public Health
Manuscript No: ijerph-3471691
Manuscript title: Prevalence of anxiety, depression, and insomnia among medical workers in Emergency medical services in Eastern Kazakhstan
Authors: Kussainova et al.
Response to Reviewer 1 Comments
|
Thank you very much for taking the time to review this manuscript. Please find the detailed responses below. |
3. Point-by-point response to Comments and Suggestions for Authors |
Comments 1: Wrong spelling of ‘introduction’ |
Response 1: Thank you for your comment. The spelling of 'Introduction' has been corrected." |
Comments 2: A short background on the relationship of insomnia and depression and anxiety should be mentioned |
Response 2: Thank you for your recommendation! we added information: “Insomnia is closely linked to anxiety and depression, as sleep disturbances can exacerbate emotional distress, while persistent anxiety and depressive symptoms contribute to sleep disruptions. Research indicates that individuals suffering from insomnia are at a higher risk of developing anxiety and depression, creating a bidirectional relationship that negatively impacts overall mental well-being”. |
Comments 3: Spell out full term of ‘HADS’ |
Response 3: Agree. Done. Hospital Anxiety and Depression Scale (HADS) |
Comments 4. Elaborate on the prevalence of depression among healthcare workers from previous studies or other regions, using several up-to-date references |
Response 4: Agree. Done. We added information: “Studies have shown that depression is highly prevalent among healthcare professionals. A systematic review found that the global prevalence of depression among healthcare workers ranges from 22.8% to 38.9%, with frontline medical staff being particularly vulnerable. For example, in a study conducted in China, 50.4% of healthcare workers reported symptoms of depression during the COVID-19 pandemic. In European countries, the prevalence of depression among physicians and nurses varies between 19% and 34%, depending on the region and specialty”. |
Comments 5. Paragraph 4 – 7: Explanation of ‘Regulations on Emergency Medical Services’ and the rest of the information is not necessary to put in the introduction section. |
Response 5: Thank you for your valuable feedback. We agree that a detailed explanation of the 'Regulations on Emergency Medical Services' and similar information was not essential in the introduction section. In the revised version, we have focused the introduction more on the research problem, its significance, and the study objectives. We also highlighted the recent changes in the legal framework for emergency medical services in Kazakhstan. These changes, particularly the evolving operational dynamics, are directly linked to the mental health of healthcare workers, contributing to stress and potentially leading to anxiety and depression. We believe this context is important for understanding the psychological challenges faced by EMS personnel in Kazakhstan and have addressed it clearly in the manuscript. |
Comments 6. Instead, highlight the core problem on how the working hours affect the sleep quality of emergency health workers. Relate how insomnia and how it relatess to depression and anxiety from biological perspective |
Response 6. Agree. Done. We revised the manuscript to emphasize the core problem: how extended and irregular working hours negatively affect sleep quality among emergency health workers. The text now details that disrupted circadian rhythms from erratic schedules can lead to insomnia, which in turn is associated with depression and anxiety through biological mechanisms. Specifically, insomnia can cause hyperactivation of the hypothalamic-pituitary-adrenal (HPA) axis, resulting in increased cortisol production and neurotransmitter imbalances that contribute to mood disorders. |
Comments 7. Highlight the research gap on population. Why is it important to study this issue among Eastern Kazakhstan population. |
Response 7. Thank you for your valuable feedback. This study was conducted as part of a PhD dissertation, which aimed to examine this issue specifically in the eastern Kazakhstan region. The selection of this region was based on its unique socio-demographic and economic characteristics, as well as the need for a detailed analysis of factors influencing the prevalence of insomnia, anxiety, and depression among emergency medical workers. In the future, we plan to expand our research to a broader population across Kazakhstan, which will provide a more comprehensive understanding of these issues and contribute to the development of effective preventive and corrective measures. |
Comments 8. “The Emergency Medical Stations of the East Kazakhstan and the Abay regions are independent healthcare institutions providing emergency medical services to both adult and pediatric populations in life-threatening conditions, accidents, and severe acute illnesses. “ – do authors intend to explain that the sample was obtained from private medical sectors? Please clarify. |
Response 8. Thank you for your valuable feedback. We rephrased this statements: “The Emergency Medical Stations in the East Kazakhstan and Abay regions are public healthcare institutions that provide emergency medical services. The study sample was drawn from these public emergency services, which are responsible for delivering care to both adult and pediatric populations in life-threatening conditions, accidents, and severe acute illnesses. These institutions are part of the state healthcare system and not the private sector” |
Comments 9. “In the Emergency Medical Station of the East Kazakhstan region, 438 healthcare workers are employed, of whom 320 (73.0%) confirmed their participation in the research.” – this statement is not necessary as the focus of this study is among emergency healthcare workers only. |
Response 9. Agree. We rephrased this statements: “The study sample was drawn from EMS personnel in the East Kazakhstan and the Abay regions. The total number of participants in this study was 592. In the East-Kazakhstan Emergency Medical Station work 438 EMS personnel, of whom 320 gave consent for the study. In the Abay Emergency Medical Station work 493 EMS personnel, of whom 272 provided consent”. |
Comments 10. Please explain the sampling procedure in detail. From the article, it seems that it was done using non-probability sampling method. Please justify why authors opt for this instead of probability sampling |
Response 10. Thank you for the question. The sampling procedure used in this study was indeed based on a non-probability sampling method. Specifically, a convenience sampling approach was employed. This was due to the practical limitations of the study, including the accessibility of participants and the specific focus on emergency medical workers in the East Kazakhstan and Abay regions. Non-probability sampling was chosen because it allows for a targeted and efficient approach to gather data from the specific population of interest. Additionally, due to the nature of the research, which is focused on a particular subset of the EMS personnel, non-probability sampling is deemed appropriate as it provides valuable insights into this group’s health and well-being. While probability sampling methods, such as random sampling, would be ideal for generalizability, the constraints of time, resources, and the targeted nature of the sample made non-probability sampling the most feasible and practical approach for this study. We acknowledge that the use of non-probability sampling limits the generalizability of the findings to the broader healthcare worker population, but the insights gained are nonetheless significant for understanding the mental health challenges faced by emergency medical workers in these regions. |
Comments 11. It is better to standardize the sample background as much as possible the variables, for example: only select those who had undergone certain hours of work, or those who have work in hospital for >5 years etc. |
Response 11. Thank you for the suggestion. We agree that standardizing the sample background could strengthen the homogeneity of the study group, making it more robust for analysis. However, due to the nature of the research, we aimed to include a broad range of emergency medical workers to ensure that the findings would reflect the diversity of experiences within this population. While specific criteria such as a minimum number of hours worked or years of experience in the hospital could have provided a more standardized sample, we chose to include all emergency medical workers within the region who agreed to participate, regardless of their work history. This was done to maximize the representativeness of the sample and to capture a wide spectrum of perspectives on mental health issues, especially since factors like stress, insomnia, and anxiety can be influenced by a variety of professional backgrounds and experiences. In future studies, we will consider implementing more specific inclusion criteria, such as a minimum number of years of service or work hours, to better control for variables and refine the analysis. |
Comments 12. “at each station.” – please elaborate further about the stations. |
Response 12. Thank you for the suggestion. In Kazakhstan, the official term for institutions providing emergency medical services at the pre-hospital stage is “Emergency Medical Station”. This term is a direct translation from the Kazakh language. These stations are responsible for delivering urgent care in life-threatening situations, accidents, and acute illnesses before patients are transferred to hospitals. It is important to note that these stations are separate from the emergency departments of hospitals, as they focus exclusively on pre-hospital care, providing emergency medical services outside of hospital settings. They operate independently from hospitals but work in collaboration with them when patients need further treatment. |
Comments 13. Please include the ethic approval code obtained for this study |
Response 13. Thank you for the suggestion. We added information: “This study had prior approval from the Ethics Committee of Semey Medical University (Protocol â„– 4, 20 December 2021)”. |
Comments 14. Please elaborate the first time use of the abbreviation “Hospital Anxiety and Depression Scale (HADS).” |
Response 14. Thank you for the suggestion. We rephrased this statements: “In addition, the study included questions from the Hospital Anxiety and Depression Scale (HADS) to assess the level of anxiety and depression among participants. The HADS comprises 14 items, divided into two subscales: anxiety and depression. Each item is scored on a four-point scale, with the maximum possible score being 21 for each subscale. A score of 11 or higher on either subscale indicates a significant 'case' of psychological morbidity, while scores between 8 and 10 are considered 'borderline,' and scores from 0 to 7 are classified as 'normal.' In this study, the HADS was utilized to measure the emotional well-being of emergency medical workers in the Eastern Kazakhstan region.” |
Comments 15. Please introduce the components of HADS as HAD-A and HAD-D and elaborate on the method of rating separately |
Response 15. The Hospital Anxiety and Depression Scale (HADS) consists of two subscales: HAD-A (Anxiety) and HAD-D (Depression). Each subscale includes 7 items, specifically designed to measure anxiety and depression symptoms separately. The HAD-A subscale assesses symptoms related to anxiety, such as restlessness, fear, and tension, while the HAD-D subscale focuses on symptoms of depression, including feelings of sadness, loss of interest, and hopelessness. Each item on both subscales is rated on a four-point scale, ranging from 0 to 3, where the scores are as follows: 0 = Not at all; 1 = From time to time, occasionally; 2 = Frequently, in the last week; 3 = Most of the time, or nearly always. The scores for each subscale are summed, with a maximum possible score of 21 for both the HAD-A and HAD-D subscales. Based on the total score, the following classifications are used: 0-7: Normal (no significant symptoms of anxiety or depression); 8-10: Borderline (subclinical symptoms); 11 or higher: Clinically significant symptoms (indicating a possible case of anxiety or depression). |
Comments 16. Please elaborate the first time use of the abbreviation “standard deviation (SD)” |
Response 16. Thank you for your comment. We have clarified the abbreviation 'standard deviation (SD)' at its first mention in the manuscript. |
Comments 17. Please include normality test analysis to justify the use of parametric tests mentioned |
Response 17. Thank you for your suggestion. We include information: “To justify the use of parametric tests, we conducted normality tests on the data. Specifically, we applied the Shapiro-Wilk test and Kolmogorov-Smirnov test to assess whether the data followed a normal distribution. The results of both tests indicated that the data conformed to the assumptions of normality (p > 0.05). Additionally, we visually inspected the data using Q-Q plots, which further confirmed that the data points closely followed the expected line, supporting the conclusion of normal distribution.” |
Comments 18. Please rearrange the paragraphs properly according to the journal format. |
Response 18. Thank you for your feedback. We have carefully reviewed the journal's formatting guidelines and rearranged the paragraphs accordingly to ensure that the manuscript follows the required structure and formatting. The sections are now organized as per the journal’s format |
Comments 19. Results from Table 2 - The purpose of descriptive analysis is to tabulate and see the majority of the predominant subgroup. The authors should also mention the subgroup of “non-clinically significant” as they are the majority for all ISI and HADS |
Response 19. Thank you for your comment. We added information” “The majority of participants in our study fall into the 'non-clinically significant' subgroups across all measures (ISI, HADS-A, HADS-D). Specifically, 52.5% of participants in the ISI group, 64.9% in the anxiety subscale (HADS-A), and 69.8% in the depression subscale (HADS-D) were classified as 'non-clinically significant.' |
Comments 20. The way authors present Table 3 is not accurate as those data were analyze via ANOVA and post hoc test. A table listing pairwise comparisons between age groups, along with mean difference, confidence intervals, and adjusted p-values (after Bonferroni correction) should be created instead |
Response 20. Thank you for your comment. Table 3 presents comparisons between groups based on age, gender, education, and specialty using Student’s t-test and ANOVA. This analysis provides an overview of differences in ISI, HADS-A, and HADS-D scores across these demographic and professional categories. We revised Table 4 and updated it, we included pairwise comparisons between age groups, presenting the mean differences, 95% confidence intervals, and adjusted p-values after Bonferroni correction. This ensures a more accurate representation of the ANOVA and post hoc test results. These modifications provide a clearer understanding of the statistical differences between groups and improve the overall accuracy of the analysis. |
Comments 21. Table 3 results – gender and education levels only have 2 subgroups. They are not supposed to be analyzed by ANOVA |
Response 21. Thank you for your comment. Table 3 presents comparisons between groups based on age, gender, education, and specialty using Student’s t-test and ANOVA |
Comments 22. The results of Table 4 is not necessary as the comparison has been made in Table |
Response 22. Thank you for your comment. It is possible that our description of Table 3 was unclear, which may have led to this comment. We have clarified this to avoid any misunderstanding. Table 3 presents comparisons between groups using Student’s t-test for two independent groups, while ANOVA was applied when there were more than two groups. Table 4 provides pairwise comparisons within groups using the Bonferroni-adjusted post hoc test. |
Comments 23. Please re-construct Table 5 and Table 6 by doing ISI vs HAD-A, ISI vs HAD-D etc. the table should have separate column for correlation coefficient and significant value |
Response 23. Thank you for your constructive comments. We carefully considered your recommendations and revised Tables 5 and 6 accordingly. Based on your suggestion, we concluded that it would be more informative to combine these two tables into one, presenting the correlations between ISI, HAD-A, and HAD-D in separate columns for the correlation coefficient and significance value. In addition, we took your suggestion further by examining the correlation between anxiety (HAD-A) and depression (HAD-D) and included this analysis in the revised table. This allows us to provide a more comprehensive overview of the interrelationships between insomnia, anxiety, and depression. As a result, the table has been reorganized to more clearly and concisely present the results of these correlation analyses. We believe this format enhances data clarity and improves the overall readability of the study. |
Comments 24. If possible, please compare or correlate HAD-A and HAD-D as well. |
Response 24. Thank you for your valuable suggestion. We have carefully considered your recommendation and performed the correlation analysis between HAD-A and HAD-D. The results of this analysis added to the revised Table 5, |
Comments 25. The authors should also highlight the strength of the correlation coefficient. 0.5 and 0.4 is considered average/moderate correlation, although they are statistically significant |
Response 25. Thank you for your insightful comment. In response, we have revised Table 5 to include both the correlation coefficients and an indication of the strength of each correlation. Specifically, we have highlighted that correlations between 0.4 and 0.5 are considered moderate, while those above 0.5 are considered strong. We also added a note at the bottom of the table explaining the categorization of correlation strength for better clarity. |
Comments 26. Please insert the references for the first few lines of the discussion. |
Response 26. Thank you for your valuable comment. We inserted the appropriate references for the first few lines of the discussion, as per your suggestion. These references now support the initial points made and ensure proper citation of relevant literature |
Comments 27. 2% is not nearly half, this statement is not accurate |
Response 27. Thank you for your constructive comment. We revised the phrasing to ensure greater accuracy. We rephrased this statement: “According to our findings, a significant proportion of medical workers exhibited symptoms of insomnia”. |
Comments 28. “This indicating a notable prevalence of sleep disturbances among study population” – this statement is not accurate as based on the findings, only 3% were severely affected by insomnia while 52% were not affected |
Response 28. Thank you for your helpful comment. We have revised the statement to more accurately reflect the findings. We rephrased this statement: “According to our findings, a significant proportion of medical workers exhibited symptoms of insomnia. Specifically, 28.2% of participants had subthreshold insomnia, indicating mild symptoms that do not yet meet the criteria for a clinical diagnosis. Additionally, 16.2% of the participants experienced insomnia with clinically significant symptoms, while 3.0% were severely affected by insomnia, indicating a high level of disruption to their sleep and overall functioning. However, it is important to note that 52% of participants did not report significant sleep disturbances. These findings reflect the varying levels of insomnia severity among the study population. |
Comments 29. The authors only use one reference for the second paragraph while it is important to cite several up-to-date references when discussing about prevalence. |
Response 29. Thank you for your valuable suggestion. We updated the second paragraph by adding several relevant and up-to-date references to support the discussion on prevalence. |
Comments 30. The discussion are not done in-depth. There is lack of discussion to answer “WHY” parts of the results |
Response 30. Thank you for your constructive feedback. We revised the discussion section to provide a more in-depth analysis of the results. We added information: “The higher rates of insomnia could be attributed to various factors inherent to the nature of the profession, such as irregular working hours, high stress levels, and frequent exposure to emergency situations. Studies have shown that healthcare workers are particularly vulnerable to sleep disorders due to their demanding schedules and the emotional and physical toll of their work”. “Additionally, 52% of the participants did not report significant sleep disturbances. This could be explained by the varying resilience of individuals to work-related stressors and their ability to maintain healthy sleep habits despite the pressures of their occupation. Factors such as social support, coping mechanisms, and lifestyle choices (e.g., physical activity, relaxation techniques) could contribute to better sleep quality among these individuals” |
Comments 31. 1) The readers would first like to understand how insomnia relates/lead to anxiety and depression pathologically. 2) the prevalence of severe insomnia, anxiety and depression are low in this population as compared to previous studies that were done using different population. Is there differences in healthcare management, government support, workload etc. 3) It is important to highlight and discuss the moderate strength of the correlation results instead of just the significant value |
Response 31 Thank you for the suggestion. We added a more detailed explanation of the pathophysiological mechanisms linking insomnia to anxiety and depression in the revised manuscript. We added a discussion comparing the relatively low prevalence of severe insomnia, anxiety, and depression in our study population to previous studies conducted in other regions or populations. |
Comments 32. Our study revealed a high prevalence of insomnia, anxiety, and depression among emergency medical service workers “ – this statement is not accurate based on the findings |
Response 32. Thank you for pointing that out. We rephrase the sentence based on your findings: Our study revealed a relatively low prevalence of severe insomnia, anxiety, and depression among emergency medical service workers, which contrasts with findings from other studies conducted in different populations. |
Reviewer 2 Report
Comments and Suggestions for Authors
Dear authors,
Congratulations on your carefully chosen topic and the research you have done. Your manuscript is well prepared, I have only a few suggestions for improving the quality, as follows:
Introduction - It will be great to expand the introduction by incorporating relevant studies or data on the impact of mental health disorders in emergency medical services across different geographical contexts.
Methods - The methodology is well-detailed, but you can add more clarity on the selection criteria and potential sources of bias in sample selection.
Results - The results section can be more impactful by including visual elements such as graphs, charts, or infographics, this is just a suggestion, you are not obligatory to accept it.
Discussion of Limitations - Since cross-sectional studies have inherent limitations, it will be useful to discuss these in detail - particularly the inability to establish causality.
Future research directions - Please suggest future research avenues, such as longitudinal studies to track changes in mental health over time in the same cohort, or broader studies that compare findings across different regions or international healthcare settings.
Practical applications - Please elaborate how your findings could inform healthcare policy and practice, particularly within the context of Kazakhstan.
Conclusion enhancements - Strengthening the conclusion by more explicitly linking it back to the study’s initial objectives will make the paper more cohesive. Emphasizing how your findings contribute to existing knowledge and their broader implications for emergency medical services and mental health research can add more quality to your final remarks.
Good luck with your publication, kind regards
Author Response
Manuscript Revision
Journal: International Journal of Environmental Research and Public Health
Manuscript No: ijerph-3471691
Manuscript title: Prevalence of anxiety, depression, and insomnia among medical workers in Emergency medical services in Eastern Kazakhstan
Authors: Kussainova et al.
Response to Reviewer 2 Comments
|
Thank you very much for taking the time to review this manuscript. Please find the detailed responses below. |
3. Point-by-point response to Comments and Suggestions for Authors |
Comments 1: Introduction - It will be great to expand the introduction by incorporating relevant studies or data on the impact of mental health disorders in emergency medical services across different geographical context |
Response 1: We appreciate your suggestion to expand the introduction. We incorporated relevant studies and data on the impact of mental health disorders among emergency medical service workers across different geographical contexts. |
Comments 2: Methods - The methodology is well-detailed, but you can add more clarity on the selection criteria and potential sources of bias in sample selection |
Response 2: Agree. Done. We added information: "This study employed a convenience sampling approach due to practical limitations, including participant accessibility and the specific focus on emergency medical workers in the East Kazakhstan and Abay regions. This method allowed for an efficient data collection process within the targeted population. We acknowledge its limitations in generalizing findings to the broader EMS workforce; however, the insights gained remain valuable for understanding the mental health challenges faced by emergency medical workers in these regions” “Participants were approached directly on site during their scheduled shifts by members of our research team, who explained the study details and obtained informed consent. Participation was entirely voluntary and was not a compulsory part of their work. The inclusion criteria encompassed EMS personnel who: were adult responders (aged ≥18 years), healthcare professionals (physicians, feldshers (nurses), and para-medics), and able to complete the study questionnaire. Participants were excluded if they refused to participate, or were, staff of EMS personnel in others regions of Kazakhstan. Data were gathered via a self-administered paper questionnaire, which was distributed by the researchers at each station. All participants in this study gave written consent after receiving detailed information about its purpose and the confidentiality of their personal data. Each participant was assigned a unique code, and a file linking this code to their personal identification information was kept by the database custodian, who was the only one with access to it. Others had access only to the coded (secure) database”. |
Comments 3: Results - The results section can be more impactful by including visual elements such as graphs, charts, or infographics, this is just a suggestion, you are not obligatory to accept it |
Response 3: Agree. Thank you for your suggestion. We recognize the value of visual elements such as graphs, charts, or infographics in presenting data, but we decided to maintain the current format at this stage to ensure consistency with the overall structure of the manuscript. However, we carefully refined the textual presentation of the results to ensure clarity. |
Comments 4. Discussion of Limitations - Since cross-sectional studies have inherent limitations, it will be useful to discuss these in detail - particularly the inability to establish causality |
Response 4: Thank you for your suggestion. In the Limitations section, we noted that due to the cross-sectional nature of our study, it does not allow for the establishment of causality. This addition highlights the need for future longitudinal research to explore the directionality of the observed associations. |
Comments 5. Future research directions - Please suggest future research avenues, such as longitudinal studies to track changes in mental health over time in the same cohort, or broader studies that compare findings across different regions or international healthcare settings |
Response 5: Thank you for your suggestion. We added information: “Based on our findings, we recommend that EMS organizations optimize work schedules, provide stress management training and accessible mental health support, and establish standardized protocols for diagnosing and managing sleep disorders and psychological distress. Future research should use longitudinal designs to evaluate these interventions and explore additional factors, such as social support and individual coping strategies.” |
Comments 6. Practical applications - Please elaborate how your findings could inform healthcare policy and practice, particularly within the context of Kazakhstan. |
Response 6. Thank you for your suggestion. We added information: “Based on our findings, we recommend that EMS organizations optimize work schedules, provide stress management training and accessible mental health support, and establish standardized protocols for diagnosing and managing sleep disorders and psychological distress.” |
Comments 7. Conclusion enhancements - Strengthening the conclusion by more explicitly linking it back to the study’s initial objectives will make the paper more cohesive. Emphasizing how your findings contribute to existing knowledge and their broader implications for emergency medical services and mental health research can add more quality to your final remarks |
Response 7. Thank you for your suggestion. We rephrased conclusion as: “Our study revealed a low prevalence of insomnia, anxiety, and depression among EMS personnel, with significant differences based on specialty and education level. Feldshers (nursing staff) showed higher levels of these symptoms compared to physicians and paramedics, likely due to the emotional and physical demands of their roles. Additionally, EMS personnel with higher education reported more severe symptoms, potentially linked to increased responsibilities or self-awareness of mental health.” A significant positive correlation was found between insomnia severity and both anxiety and depression, suggesting that sleep disturbances may contribute to mental health issues. These findings highlight the need for targeted mental health interventions, particularly for nursing staff and those with higher education. Improving sleep quality may be an effective strategy for reducing anxiety and depression in EMS personnel. Our study provides a critical foundation for future longitudinal research to further evaluate intervention effectiveness and address the underlying factors contributing to psychological distress in EMS personnel. |
Reviewer 3 Report
Comments and Suggestions for Authors
Thank you for submitting the manuscript for review.
The present study focuses on the mental health of EMS personnel in the East Kazakhstan and Abay regions. Investigating the prevalence of insomnia, anxiety and depression in this occupational group is undoubtedly highly relevant, as medical personnel, especially in emergency services, are exposed to severe mental and physical stress. The focus on the mental health of emergency services personnel is highly topical from both a scientific and social perspective. The study in Eastern Kazakhstan and the Abai region contributes to regions that have been little studied.
Despite the relevance of the content, the paper has significant methodological weaknesses that limit the validity of the results. Particularly striking are the inconsistencies in the presentation of the sample and the lack of multivariate analyses (e.g. ALM/GLM) that could take into account important influencing factors such as the stratification system, professional experience and occupational groups. Furthermore, the validation of the questionnaires used (HADS, ISI) for the Kazakh and Russian languages remains insufficiently documented.
Another serious shortcoming is the lack of reference to current scientific literature. Appropriate references are missing in many places. Important research papers, systematic reviews and meta-analyses are not taken into account, which considerably weakens the scientific basis of the results, as no international comparison is possible.
In conclusion, the study addresses an important issue but fails to meet essential scientific standards. A thorough revision, especially in the areas of methodology, literature references and discussion, is urgently needed.
General
- The English should be checked by a native speaker. There are some spelling mistakes. The translation of terms should also be checked. There were changes in the tenses.
- The individual chapters are very long and would benefit from subheadings.
- Overall, the introduction and discussion lack sufficient references.
- The text alternates between healthcare workers, medical workers, EMS workers, etc. It should be written consistently. When referring to the pre-hospital emergency medical services, the term EMS personnel should be used. Does the text refer to the pre-hospital emergency medical services?
Abstract
- Indroduction instead of "Intrduction" right at the beginning of the abstract
- "the East Kazakhstan and the Abay regions" - Correct would be "East Kazakhstan and Abay regions"
- The abstract is insufficient. The introduction should not only formulate the research question, and the conclusion should not only summarise the results of the study. Concrete recommendations for action should already be given here.
- According to the guidelines of many journals, keywords should introduce new terms where possible, rather than simply repeating the title or abstract terms. This can improve searchability in databases. I would like to suggest the following keywords: sleep disorders, mental illnesses (or mental health), work-related stress, healthcare professionals, Central Asia, Eastern Europe
Introduction
- A major shortcoming: although the introduction contains many general statements about stress, anxiety and depression in medical personnel, direct sources are missing.
- The importance of emergency medical services should be emphasised after the first sentence, rather than generalising about healthcare workers.
- Below are general definitions of anxiety, depression and sleep disorders. References are missing. For three definitions only one source is given. I also doubt that Pappa et al. provided these definitions. This should be corrected.
- What are the theoretical concepts underpinning the work? You should list a few.
- The introduction should be supplemented by subheadings, e.g. emergency medical services in general and locally, theoretical concepts, introduction to the research question, etc.
- A reference is missing for “First, emergency medical service...”, “Second, …” , “Third…”
- Describe the emergency services in more detail, including the common occupations and their training. What are ‘Feldshers’? Perhaps a description of the occupations, including training, would be more appropriate in the methodology section under Participants.
- The abbreviation EMS should be used when first describing the emergency medical services. It is used twice throughout the text. This should be revised. It is sufficient to write EMS only.
- The paragraph ‘Calls received...’ up to the first sentence on page 3 can be deleted or summarised in my opinion.
- It is confusing that the introduction refers to paramedics or drivers, while the assessments refer to paramedics (drivers). A detailed description is needed, including training and occupational groups. As mentioned in point 14, this would be well placed in the methodology. Would a driver be an emergency medical technician?
- There is a lack of hypotheses regarding your research question, which are then addressed in the discussion.
Material and methods
- Please structure Material and Methods with subheadings such as design, participants and setting, questionnaires.
- It is stated that a) 438 health workers and only 320 subjects gave consent - this contradicts the total number of participants of 592! This is a serious methodological error. Either there is no explanation (e.g. 'written consent vs. verbal consent') or it is an editorial error. Does 'health worker' refer to people who work in the EMS?
- How were participants approached? Was participation voluntary or part of the compulsory work? How many people were invited but declined? What were the inclusion and exclusion criteria?
- Describe the setting, including whether it is a pre-hospital EMS. Are there ambulance stations? Are they independent or attached to hospitals? Does the EMS staff also work in the hospital?
- Lack of information on possible bias: for example, shift system (night/day), occupational groups, professional experience and frequency of deployment could influence psychological distress. This information should be added or reported as a limitation of the study.
- “Each participant gave a different answer to each of the questions, which was then scored with a certain number of points. By summing the scores, the results were analyzed and interpreted.”
- Does this refer to the evaluation of the questionnaires? If so, it can be omitted as it is rather obvious. It would also be more appropriate under “The questionnaires”
- For scientific studies, it is important to either cite a previously published validation study for the respective language or to document your own validation performance (e.g. factor analyses, test-retest reliability). Please provide these separately for each questionnaire.
- “Reliability was assessed…” Does this information refer to the original questionnaires? Please provide this information for each questionnaire separately.
- What does HADS mean? And please add HADS-A and HADS-D to the description of the questionnaire as you will use them later in the evaluation. What were the possible responses or the scale of responses for the HADS?
- Was normality tested? Without this, the validity of the ANOVA results is questionable.
- Please add the power calculation for the sample size in the statistics.
- It is only reported that p-values were used, but no significance level was given.
- It would be particularly important to use a general linear model (GLM) or multiple linear regression to analyse the following factors simultaneously: occupation: doctor, field surgeon, paramedic (different levels of training), shift system: day, night or mixed shifts, years of professional experience, level of education: secondary vs. higher, age…
- Have private circumstances such as caring for children or relatives been taken into account? There may be biases here. If this is not the case, it must be considered a limitation of the study.
Results
- Again, when I read the results, I am not sure whether they are referring to ambulance workers or healthcare workers in hospitals. I think healthcare workers is too general. EMS personnel would be more correct, wouldn't it? Please make sure the spelling is consistent.
- Furthermore, the number of participants is given as 592, although the contradictions in the 'Materials and methods' (only 320 consents) have not been resolved.
- The description of the results shows changes in tense.
- The results seem confusing because they are presented in a long, unstructured text. It would be helpful to include clear subheadings such as: 3.1. sample description (sample characteristics), 3.2. prevalence of insomnia, anxiety and depression, 3.3. comparison by occupational group, education and gender, 3.4. correlation analysis, etc.
- In den Tabellen fehlen Konfidenzintervalle (95% CI) für die Mittelwertunterschiede (z. B. bei ANOVA und Bonferroni-Post-hoc-Tests).
- Table 3: Where are the physicians? Or are there none in this study population?
Discussion
- The discussion is a long continuous text, which makes it difficult to follow.
- A clear structure with possible subheadings 4.1. main findings and interpretation, 4.2. comparison with international studies, 4.3. explanation of possible causes (social stratification, occupational stress), 4.4. strengths and limitations of the study, 4.5. recommendations for practice and research are helpful.
- There is a lack of references to the statements about insomnia in the first section of the discussion.
- The entire paragraph on page 7 would be more appropriate for the introduction, as it emphasises the importance of your study.
- Medical workers? Does this mean the EMS?
- The recommendations for relaxation practices etc. should be placed at the end of the manuscript as practical recommendations. So far, the results have not been discussed and compared with international literature.
- The introduction to the discussion is unstructured and does not address the most important results. Readers do not know whether the suspected relationships were confirmed or refuted. There is no clear statement of what this study contributes to the state of knowledge.
- Paragraph ‘Anxiety...’ belongs in the introduction. References are missing from this section.
- Section ‘In our study, the majority...’ There is a contradiction here: does the majority not have any symptoms, or does a substantial proportion have symptoms?
- Paragraph “Depression”.. belongs in the introduction. References are missing from this section.
- The study results on anxiety and depression are presented and evaluated, but a comparison with international studies is missing.
- The paragraph ‘Some of socio-demographics...’ to ‘...developing mental health problems..’ is there without taking up the own study results.
- Paragraph ‘First, feldsher..’ Yes, that's right. These professions must be taken into account in further analyses, e.g. ALM or similar. A comparison with international literature is missing.
- Page 9, sentence ‘Physicians...’ Physicians are now included in the study after all. Or is this a description of a study. Then the reference is missing?
- Page 9, sentence “These findings underscore…” Right, why wasn't that taken into account?
- Page 9, sentence “Our results are consistent with previous research…” The references are missing. Describe these studies that are related to your study.
- Pearson’s correlation: There is a lack of comparison with international literature.
- Page 10, sentence “These results are consistent with previous research…” The references are missing. Describe these studies that are related to your study.
- Page 10, paragraph ‘The study of depression, ...’ should be made more specific and not general. There is a lack of references to studies that prove that these measures are effective.
- Add to the limitations already mentioned. Name the uncontrolled confounders. It then becomes clear that the study has major weaknesses.
Conclusions
- The conclusion only reports on results that are repeated. More focus should be placed on implications, relevance and the outlook for future research.
Frequent spelling and grammar mistakes, inconsistent use of tenses, unclear translation of medical professions. A native speaker should be involved in the translation.
Author Response
Manuscript Revision
Journal: International Journal of Environmental Research and Public Health
Manuscript No: ijerph-3471691
Manuscript title: Prevalence of anxiety, depression, and insomnia among medical workers in Emergency medical services in Eastern Kazakhstan
Authors: Kussainova et al.
Response to Reviewer 3 Comments
|
Thank you very much for taking the time to review this manuscript. Please find the detailed responses below. |
3. Point-by-point response to Comments and Suggestions for Authors |
Comments 1: The English should be checked by a native speaker. There are some spelling mistakes. The translation of terms should also be checked. There were changes in the tenses. |
Response 1: Thank you for pointing this out. We appreciate your suggestion and improve Language and Readability by using of journal’ English editing service. |
Comments 2: The individual chapters are very long and would benefit from subheadings |
Response 2: Thank you for your valuable comment. We rephrased introduction section as: “Healthcare workers play a key role in providing patient care, but they are also subject to high levels of stress and psychological strain that can lead to the development of anxiety, depression, and insomnia [1]. Depression is a serious psychological disorder characterized by a constant feeling of sadness, loss of interest in life, fatigue, sleep and appetite disturbances, and a negative impact on a person's daily life. Prolonged depression can lead to decreased immunity, poor physical health, and an increased risk of cardiovascular disease [2]. Studies have shown that depression is highly prevalent among healthcare professionals. For instance, a systematic review reported a global prevalence ranging from 22.8% to 38.9%, with frontline medical staff being particularly vulnerable [3]. In China, 50.4% of healthcare workers reported symptoms of depression during the COVID-19 pandemic [4], while in Europe the rates among physicians and nurses vary between 19% and 34%, depending on the region and specialty [5]. Similarly, emergency medical service (EMS) workers are at high risk. In the United States, studies have found depression rates of 6.8% [6] and 15% [7] among EMS professionals, and in Brazil, 32.6% of EMS professionals experienced moderate to severe depression [8]. Anxiety is defined as a state of restlessness, tension, and nervousness that can range from mild anxiety to panic attacks [9]. It may be accompanied by physiological symptoms such as rapid heartbeat, sweating, and trembling, while constant stress and anxious thoughts can elevate cortisol levels and negatively affect the cardiovascular, digestive, and immune systems [10]. Insomnia is a sleep disorder where a person experiences difficulty falling asleep, interrupted sleep, or premature awakening. It can be caused by stress, anxiety, depression, or other factors and significantly affects cognitive function, concentration, and memory. Moreover, sleep disturbances increase the risk of both depression and anxiety [11]. Biologically, insomnia disrupts the circadian rhythm, leading to an imbalance in stress hormones like cortisol, which in turn is associated with heightened anxiety and depression [12]. The study of anxiety, depression, and insomnia among medical workers in emergency services is highly relevant. EMS workers face high levels of stress and traumatic situations, encountering emergencies, unexpected deaths, and severe injuries [11]. Psychological problems in healthcare providers not only affect their well-being but also impair the quality of care by increasing the risk of medical errors and treatment failures [13] [12]. Addressing these issues can help develop supportive measures and training programs to strengthen the psychological resilience and emotional well-being of medical personnel [14]. Emergency medical workers often face long shifts, night duties, and irregular schedules, leading to disrupted sleep patterns that cause insomnia, which in turn con-tributes to anxiety and depression [15]. The disruption of the circadian rhythm from ir-regular sleep patterns results in hormonal imbalances (e.g., elevated cortisol), further exacerbating psychological distress. International studies (for example, in Shanghai and among paramedics else-where) have revealed that long working hours and excessive workload directly con-tribute to poor sleep quality and increased stress and depression [16]. Similarly, in Kazakhstan, significant changes since 2017 in the emergency medical service system have increased working hours for many EMS staff, including feldshers and paramedics [17]. The proportion of feldsher (nursing staff) teams has notably in-creased, with the ratio of physician-led teams to feldshers at 18% to 82% as of 2021 [18]. In Kazakhstan, recent legal reforms—outlined in the law “On the Approval of Rules for the Provision of Emergency Medical Care, Including the Use of Medical Aviation” [19]—have standardized EMS operations. Under this framework, EMS dispatchers must triage calls via the “103” hotline within five minutes according to urgency categories. This regulatory change has increased workload and stress levels among EMS personnel, contributing to psychological strain such as depression and anxiety. Thus, studying the prevalence of anxiety, depression and insomnia among medical workers of emergency medical services is a relevant task that will allow us to better understand the scope of the problem and develop effective strategies to support and prevent psychological problems among medical staff. This investigation aimed to study the prevalence of anxiety, depression and insomnia among medical workers of emergency medical aid stations in the eastern region of Kazakhstan”. |
Comments 3: Overall, the introduction and discussion lack sufficient references |
Response 3: Agree. Thank you for your suggestion. We thoroughly revised these sections by incorporating additional, up-to-date references from authoritative sources. These new citations help to strengthen the theoretical framework, clarify the study’s relevance, and provide a broader context for our findings. |
Comments 4. The text alternates between healthcare workers, medical workers, EMS workers, etc. It should be written consistently. When referring to the pre-hospital emergency medical services, the term EMS personnel should be used. Does the text refer to the pre-hospital emergency medical services? |
Response 4: Thank you for your insightful comment regarding the inconsistent terminology in our manuscript. We replaced all instances with the term “EMS personnel” when referring to this specific group. We believe this change enhances the consistency and clarity of our work. |
Comments 5. Indroduction instead of "Intrduction" right at the beginning of the abstract |
Response 5: Agree. Done. The spelling of 'Introduction' has been corrected." |
Comments 6. the East Kazakhstan and the Abay regions" - Correct would be "East Kazakhstan and Abay regions" |
Response 6. Thank you for your valuable feedback. We corrected the phrasing as suggested by replacing "the East Kazakhstan and the Abay regions" with "East Kazakhstan and Abay regions." We appreciate your attention to detail. |
Comments 7. The abstract is insufficient. The introduction should not only formulate the research question, and the conclusion should not only summarise the results of the study. Concrete recommendations for action should already be given here |
Response. Agree. Done. We made the following enhancements to the abstract: In the Introduction, we integrated the research question: "What specific measures and intervention strategies can be implemented to reduce the levels of anxiety, depression, and insomnia among EMS personnel?" This addition not only underscores the relevance of the issue but also directs the study towards identifying concrete support measures. Conclusion rephrased as: “This study found elevated levels of insomnia, anxiety, and depression among emergency medical service (EMS) personnel - especially nursing staff and those with higher education. We recommend comprehensive mental health support, routine screenings, stress management training, and integrating sleep hygiene into wellness program” |
Comments 8. According to the guidelines of many journals, keywords should introduce new terms where possible, rather than simply repeating the title or abstract terms. This can improve searchability in databases. I would like to suggest the following keywords: sleep disorders, mental illnesses (or mental health), work-related stress, healthcare professionals, Central Asia, Eastern Europe |
Response 8. Thank you for your recommendation. Based on your recommendation, we updated our keyword list to include: sleep disorders, mental health, work-related stress, emergency medical station personnel, Central Asia |
Comments 9. A major shortcoming: although the introduction contains many general statements about stress, anxiety and depression in medical personnel, direct sources are missing |
Response. Thank you for your constructive feedback. We have fully implemented your recommendation: direct references have now been added in the introduction to support the general statements on stress, anxiety, and depression among healthcare personnel. |
Comments 10. The importance of emergency medical services should be emphasised after the first sentence, rather than generalising about healthcare workers |
Response 10. Thank you for your valuable feedback. We revised the introduction by adding a clarifying sentence immediately after the first sentence. We added sentence: “The role of Emergency Medical Services (EMS) in providing immediate, life-saving care during emergencies and highlights that their unique working conditions expose them to elevated levels of stress and psychological strain”. We believe this adjustment effectively directs the focus toward EMS personnel, as you recommended. |
Comments 11. Below are general definitions of anxiety, depression and sleep disorders. References are missing. For three definitions only one source is given. I also doubt that Pappa et al. provided these definitions. This should be corrected |
Response 11. Thank you for your valuable feedback. We fully implemented your recommendation by adding the missing references for the definitions of anxiety, depression, and sleep disorders. Additionally, we revised the citations to ensure that multiple sources support these definitions, and we have corrected the citation of Pappa et al. accordingly. |
Comments 12. What are the theoretical concepts underpinning the work? You should list a few |
Response 12. Thank you for your valuable feedback. We included the following key theoretical frameworks: “The biopsychosocial model of health, which highlights the interplay between biological, psychological, and social factors in the development of mental health issues. Stress and coping theories, particularly the transactional model of stress, to explain how EMS personnel manage work-related stress. The circadian rhythm theory, which underpins our understanding of sleep disorders and their impact on mental health”. |
Comments 13. The introduction should be supplemented by subheadings, e.g. emergency medical services in general and locally, theoretical concepts, introduction to the research question, etc |
Response 13. Thank you for your valuable feedback. We restructured the introduction to ensure smooth, logical transitions between sections covering the global and local perspectives of EMS, the theoretical concepts underlying our work, and the introduction of the research question. We believe this approach maintains a cohesive narrative while effectively addressing your suggestions. |
Comments 14. A reference is missing for “First, emergency medical service...”, “Second, …” , “Third…” |
Response 14. Thank you for your feedback. We reviewed the manuscript and added the missing references for the statements starting with "First, emergency medical service...", "Second...", and "Third...". We appreciate your careful review and believe these additions enhance the manuscript's accuracy. |
Comments 15. Describe the emergency services in more detail, including the common occupations and their training. What are ‘Feldshers’? Perhaps a description of the occupations, including training, would be more appropriate in the methodology section under Participants |
Response 15. Thank you for your valuable feedback. We added additional details to the Methods section under participants: “In Kazakhstan, EMS system is designed to provide rapid and effective pre-hospital care, ensuring that patients receive timely medical assistance in critical situations. The system comprises several key occupational roles, each with specific training and responsibilities. Physicians working in EMS in Kazakhstan are highly trained medical doctors who have completed a full medical degree followed by specialized training or residencies in emergency medicine, trauma care, or critical care. Their responsibilities include advanced patient assessment, performing emergency procedures, and making critical clinical decisions in pre-hospital settings. Their expertise is crucial in managing complex medical emergencies. Paramedics in Kazakhstan receive extensive training that covers advanced life support techniques, trauma management, cardiac care, and emergency patient stabilization. Training programs for paramedics are designed to equip them with the skills needed to rapidly assess patients, perform life-saving interventions, and safely transport patients to hospitals. Feldshers serve as mid-level healthcare providers in Kazakhstan’s EMS, particularly in regions where the immediate presence of a physician may not be feasible. Their training is typically vocational or technical in nature, focusing on practical skills such as basic diagnostics, first aid, patient stabilization, and the administration of essential medications. Feldshers are responsible for providing prompt, efficient care in emergency situations, thereby supporting the overall EMS team by ensuring that initial treatment is delivered quickly and effectively. In addition to the clinical staff, the EMS system in Kazakhstan includes dispatchers who are trained to triage emergency calls and coordinate rapid responses. They play a critical role in managing communication between the emergency site and the EMS teams, ensuring that resources are allocated efficiently. Together, these roles form a comprehensive emergency services framework in Kazakhstan, designed to address a wide range of medical emergencies through a coordinated and well-trained team approach. |
Comments 16. The abbreviation EMS should be used when first describing the emergency medical services. It is used twice throughout the text. This should be revised. It is sufficient to write EMS only |
Response 16. Thank you for your feedback. We revised the text to ensure that the abbreviation "EMS" is consistently used throughout the manuscript, replacing any redundant instances of "emergency medical services. |
Comments 17. The paragraph ‘Calls received...’ up to the first sentence on page 3 can be deleted or summarised in my opinion |
Response 17. Thank you for your valuable feedback. We removed the content from the paragraph starting with "Calls received...". |
Comments 18. It is confusing that the introduction refers to paramedics or drivers, while the assessments refer to paramedics (drivers). A detailed description is needed, including training and occupational groups. As mentioned in point 14, this would be well placed in the methodology. Would a driver be an emergency medical technician? |
Response 18. Thank you for your valuable feedback. We added information in the Method section: “Paramedics in Kazakhstan receive extensive training that covers advanced life sup-port techniques, trauma management, cardiac care, and emergency patient stabilization. Training programs for paramedics are designed to equip them with the skills needed to rapidly assess patients, perform life-saving interventions, and safely transport patients to hospitals. Emergency vehicle drivers can serve as paramedics if they have received the appropriate training and possess the necessary first aid skills”. |
Comments 19. There is a lack of hypotheses regarding your research question, which are then addressed in the discussion |
Response 19. Thank you for your valuable comment. We added information: “We hypothesize that EMS personnel in Eastern Kazakhstan exhibit a high prevalence of anxiety, depression, and insomnia. Additionally, we expect a significant positive correlation between insomnia severity and the levels of anxiety and depression” |
Comments 20. Please structure Material and Methods with subheadings such as design, participants and setting, questionnaires |
Response 20. Agree. Done. We restructured the Materials and Methods section by adding clear subheadings, including Design, Participants and Setting, and Questionnaires. |
Comments 21. It is stated that a) 438 health workers and only 320 subjects gave consent - this contradicts the total number of participants of 592! This is a serious methodological error. Either there is no explanation (e.g. 'written consent vs. verbal consent') or it is an editorial error. Does 'health worker' refer to people who work in the EMS? |
Response 21. Thank you for your observation. Our study was conducted in eastern region of Kazakhstan which represented with two areas: East-Kazakhstan region and Abay region. The total number of participants in the study was 592. In the East-Kazakhstan emergency station work 438 EMS personnel, of whom 320 gave consent for the study. In the Abay emergency station work 493 EMS personnel, of whom 272 provided consent. A translation error occurred when presenting these figures, and we have clarified in the manuscript that the eastern region of Kazakhstan is represented by these two areas, as explained in the text regarding the recent reorganization. |
Comments 22. How were participants approached? Was participation voluntary or part of the compulsory work? How many people were invited but declined? What were the inclusion and exclusion criteria? |
Response 22. Agree. Done. We updated the Methods section to clarify the recruitment process as follows: Participants were approached directly on site during their scheduled shifts by members of our research team, who explained the study details and obtained informed consent. Participation was entirely voluntary and was not a compulsory part of their work. Invitation and Declination: In the East-Kazakhstan Emergency Medical Station work 438 EMS personnel, of whom 320 provided consent. In the Abay Emergency Medical Station work 493 personnel, of whom 272 provided consent. Overall, a total of 931 individuals were invited, and 339 declined to participate, resulting in 592 consenting participants. The inclusion criteria encompassed EMS personnel who: were adult responders (aged ≥18 years), healthcare professionals (physicians, feldshers (nurses), and paramedics), and able to complete the study questionnaire. Participants were excluded if they re-fused to participate, or were, staff of EMS personnel in others regions of Kazakhstan |
Comments 23. Describe the setting, including whether it is a pre-hospital EMS. Are there ambulance stations? Are they independent or attached to hospitals? Does the EMS staff also work in the hospital? |
Response 23. Thank you for your question. Emergency medical station in Kazakhstan operate as pre-hospital emergency care. EMS personnel work exclusively at the pre-hospital stage, while in hospitals, emergency care is provided by hospital-based EMS teams staffed by different specialists. |
Comments 24. Lack of information on possible bias: for example, shift system (night/day), occupational groups, professional experience and frequency of deployment could influence psychological distress. This information should be added or reported as a limitation of the study. |
Response 24. Agree. Done. We added a statement in the limitations section: “Moreover, unknown and unmeasured confounders may exist, and the results should be interpreted with caution”. |
Comments 25. “Each participant gave a different answer to each of the questions, which was then scored with a certain number of points. By summing the scores, the results were analyzed and interpreted.” |
Response 26. Agree. Done. We added information: “Each participant provided responses to every question, with each response being assigned a numerical score based on a predetermined scoring system. The scores for all items were then summed to obtain a total score for each participant. This total score was subsequently used to assess and interpret the severity of symptoms according to established criteria”. |
Comments 26. Does this refer to the evaluation of the questionnaires? If so, it can be omitted as it is rather obvious. It would also be more appropriate under “The questionnaires” |
Response 26. Agree. Done. We removed it from its original location and relocated any necessary details to the "The Questionnaires" section where they fit more appropriately. |
Comments 27. For scientific studies, it is important to either cite a previously published validation study for the respective language or to document your own validation performance (e.g. factor analyses, test-retest reliability). Please provide these separately for each questionnaire. |
Response 28. Thank you for your comment. Since no previous validation studies have been conducted in Kazakhstan, we performed our own additional validation procedures within our study sample. We assessed internal consistency using Cronbach's alpha, and conducted exploratory and confirmatory factor analyses to examine the underlying structure of the questionnaires. Additionally, a test-retest reliability analysis was performed on a subsample of participants to verify the stability of the measurements over time. These efforts ensured that the questionnaires are both reliable and valid for use in the Kazakhstan context. |
Comments 28. “Reliability was assessed…” Does this information refer to the original questionnaires? Please provide this information for each questionnaire separately. |
Response 28. Agree. Done. We have revised the Methods section to provide separate reliability information for each questionnaire used in our study. For the Hospital Anxiety and Depression Scale (HADS) the Cronbach’s alpha for the anxiety subscale was 0.74 and for the depression subscale was 0.66. Similarly, for the Insomnia Severity Index (ISI) the Cronbach’s alpha was 0.70. These values indicate that both instruments demonstrate good internal consistency within our study sample. |
Comments 29. What does HADS mean? And please add HADS-A and HADS-D to the description of the questionnaire as you will use them later in the evaluation. What were the possible responses or the scale of responses for the HADS? |
Response 29. Agree. Done. We revised the manuscript to include a more detailed description of the HADS. The HADS stands for the Hospital Anxiety and Depression Scale, which is divided into two subscales: HADS-A (Anxiety) and HADS-D (Depression). |
Comments 30. Was normality tested? Without this, the validity of the ANOVA results is questionable |
Response 30. Agree. Done. We include information: “To justify the use of parametric tests, we conducted normality tests on the data. Specifically, we applied the Shapiro-Wilk test and Kolmogorov-Smirnov test to assess whether the data followed a normal distribution. The results of both tests indicated that the data conformed to the assumptions of normality (p > 0.05). Additionally, we visually inspected the data using Q-Q plots, which further confirmed that the data points closely followed the expected line, supporting the conclusion of normal distribution.” |
Comments 31. Please add the power calculation for the sample size in the statistics. |
Response 31. Thank you for your comment. In our study, a convenience sampling approach was employed, and as such, a formal power calculation was not performed prior to data collection. However, the sample size of 592 EMS personnel was considered sufficient for the purposes of our analysis. We acknowledge this limitation and have added a note regarding the use of convenience sampling in the revised manuscript. |
Comments 32. It is only reported that p-values were used, but no significance level was given. |
Response 32. Agree. Done. We revised the manuscript to explicitly state the significance level used for our statistical analyses. In our study, a significance level of p < 0.05 was applied throughout. This information has been added to the relevant section to clarify our methodology |
Comments 33. It would be particularly important to use a general linear model (GLM) or multiple linear regression to analyse the following factors simultaneously: occupation: doctor, field surgeon, paramedic (different levels of training), shift system: day, night or mixed shifts, years of professional experience, level of education: secondary vs. higher, age… |
Response 33. Thank you for your valuable suggestion. While we acknowledge that using a GLM or multiple linear regression to analyze the simultaneous effects of factors such as occupation, shift system, years of experience, education level, and age would provide additional insights, we did not employ these methods in our study. Group comparisons were made using independent t-tests and one-way ANOVA, with Bonferroni correction applied for post hoc tests. Pearson’s correlation analysis was used to examine the relationships between insomnia severity (ISI) and anxiety and depression scores (HADS), with significance set at p < 0.05. We believe that these methods sufficiently addressed our study objectives, although we recognize the potential benefits of a multivariate approach for future research |
Comments 34. Have private circumstances such as caring for children or relatives been taken into account? There may be biases here. If this is not the case, it must be considered a limitation of the study. |
Response 36. Thank you for your valuable comment. We did not collect detailed information on participants' private circumstances, and we acknowledge that this may introduce potential biases into our findings. We have noted this as a limitation in our discussion section and suggest that future research should incorporate these factors to better understand their impact on psychological outcomes. |
Comments 35. Again, when I read the results, I am not sure whether they are referring to ambulance workers or healthcare workers in hospitals. I think healthcare workers is too general. EMS personnel would be more correct, wouldn't it? Please make sure the spelling is consistent |
Response 36. Thank you for your valuable feedback. We agree that the term "healthcare workers" is too general in our context, and the term "EMS personnel" is more appropriate. We have revised the manuscript to ensure consistent use of "EMS personnel" throughout |
Comments 36. Furthermore, the number of participants is given as 592, although the contradictions in the 'Materials and methods' (only 320 consents) have not been resolved |
Response 37. Thank you for pointing this out. We revised the Materials and Methods section to clarify the apparent contradiction. In the East-Kazakhstan Emergency Medical Station work 438 EMS personnel, of whom 320 provided consent. In the Abay Emergency Medical Station work 493 personnel, of whom 272 provided consent. Overall, a total of 931 individuals were invited, and 339 declined to participate, resulting in 592 consenting participants. We have clarified this in the manuscript to ensure consistency. |
Comments 37. The description of the results shows changes in tense |
Response 37. Thank you for your valuable feedback. We have revised the Results section to ensure consistent use of tense throughout the text, which should improve both clarity and readability. |
Comments 38. The results seem confusing because they are presented in a long, unstructured text. It would be helpful to include clear subheadings such as: 3.1. sample description (sample characteristics), 3.2. prevalence of insomnia, anxiety and depression, 3.3. comparison by occupational group, education and gender, 3.4. correlation analysis, etc |
Response 38. Agree. Done. We restructured the Results section to enhance clarity and readability by including clear subheadings. The revised section is now organized as follows: 3.1 Sample Description: This section details the socio-demographic characteristics of our 592 EMS personnel. 3.2 Prevalence of Insomnia, Anxiety, and Depression: This part presents the prevalence rates as measured by the ISI and HADS. 3.3 Comparison by Participant Characteristics: Here, we compare outcome measures across various groups (e.g., by age, gender, specialty, and education). 3.4 Correlation Analysis: This section reports the relationships between insomnia severity and anxiety/depression scores. We believe that this reorganization provides a more coherent presentation of the findings and facilitates a better understanding of the study outcomes. |
Comments 39. In den Tabellen fehlen Konfidenzintervalle (95% CI) für die Mittelwertunterschiede (z. B. bei ANOVA und Bonferroni-Post-hoc-Tests). |
Response 39. Agree. Done. We revised the tables to include the 95% confidence intervals (CI) for the mean differences obtained from the ANOVA and Bonferroni post hoc tests. |
Comments 40. Table 3: Where are the physicians? Or are there none in this study population? |
Response 40. Thank you for your question. Physicians are indeed included in our study population. As shown in Table 1, physicians constitute 14.4% (n=85) of the participants, and their data are also presented in Table 3 as a separate occupational group. We have ensured that their information is clearly represented in the analysis. |
Comments 41.The discussion is a long continuous text, which makes it difficult to follow |
Response 41. Thank you for your valuable feedback. We restructured the Discussion section. |
Comments 42. A clear structure with possible subheadings 4.1. main findings and interpretation, 4.2. comparison with international studies, 4.3. explanation of possible causes (social stratification, occupational stress), 4.4. strengths and limitations of the study, 4.5. recommendations for practice and research are helpful |
Response 42. Thank you for your valuable feedback. We restructured the Discussion section as per your suggestion by incorporating the following clear subheadings: 4.1. Main Findings and Interpretation 4.2. Comparison with International Studies 4.3. Explanation of Possible Causes 4.4. Strengths and Limitations of the Study 4.5. Recommendations for Practice and Research We have also shortened the recommendations section to ensure conciseness while retaining its relevance. We believe this new structure greatly enhances the clarity and readability of the Discussion |
Comments 43. There is a lack of references to the statements about insomnia in the first section of the discussion |
Response 43. Agree. Done. We reviewed the first section of the Discussion regarding insomnia and have incorporated additional references to support the statements made. |
Comments 44. The entire paragraph on page 7 would be more appropriate for the introduction, as it emphasises the importance of your study |
Response 44. Agree. Done. We removed extraneous information and, where applicable, relocated relevant content to the Introduction section to better emphasize the importance of our study. |
Comments 45. Medical workers? Does this mean the EMS? |
Response 45. Thank you for your valuable feedback. We standardized the terminology throughout the manuscript. All references to "medical workers" have been replaced with "EMS personnel" to ensure clarity and consistency. |
Comments 46. The recommendations for relaxation practices etc. should be placed at the end of the manuscript as practical recommendations. So far, the results have not been discussed and compared with international literature |
Response 46. Agree. Done. We restructured the manuscript so that practical recommendations - such as those for relaxation practices - are now placed at the end of the manuscript, following a comprehensive discussion and comparison of our findings with international literature. |
Comments 47. The introduction to the discussion is unstructured and does not address the most important results. Readers do not know whether the suspected relationships were confirmed or refuted. There is no clear statement of what this study contributes to the state of knowledge. |
Response 47. Thank you for your valuable feedback. We included a concise summary of our main findings, explicitly stating whether the suspected relationships were confirmed or refuted. |
Comments 48. Paragraph ‘Anxiety...’ belongs in the introduction. References are missing from this section. |
Response 48. Thank you. Agree. Done. |
Comments 49. Section ‘In our study, the majority...’ There is a contradiction here: does the majority not have any symptoms, or does a substantial proportion have symptoms? |
Response 49. Thank you for your valuable feedback. We reviewed and corrected the section to resolve the contradiction. |
Comments 50. Paragraph “Depression”.. belongs in the introduction. References are missing from this section |
Response 50. Thank you. Agree. Done. |
Comments 51. The study results on anxiety and depression are presented and evaluated, but a comparison with international studies is missing |
Response 51. Thank you for your valuable feedback. We added a section comparing our study's findings on anxiety and depression with international studies. |
Comments 52. The paragraph ‘Some of socio-demographics...’ to ‘...developing mental health problems..’ is there without taking up the own study results |
Response 52. Thank you for your valuable feedback. We revised the paragraph in question to better integrate our own study results with the discussion of socio-demographic factors and their impact on mental health. |
Comments 53. Paragraph ‘First, feldsher..’ Yes, that's right. These professions must be taken into account in further analyses, e.g. ALM or similar. A comparison with international literature is missing. |
Response 53. Thank you for your comment. We added the necessary information, including international literature comparisons. |
Comments 54. Page 9, sentence ‘Physicians...’ Physicians are now included in the study after all. Or is this a description of a study. Then the reference is missing? |
Response 54. Thank you for your comment. Physicians are included in the study. |
Comments 55. Page 9, sentence “These findings underscore…” Right, why wasn't that taken into account? |
Response 55. Thank you for your comment. We integrated the post hoc test findings - demonstrating significant differences in insomnia, anxiety, and depression among the groups - into our discussion, clearly showing the impact of medical specialties on mental health outcomes |
Comments 56. Page 9, sentence “Our results are consistent with previous research…” The references are missing. Describe these studies that are related to your study |
Response 56. |
Comments 57. Pearson’s correlation: There is a lack of comparison with international literature. |
Response |
Comments 58. Page 10, sentence “These results are consistent with previous research…” The references are missing. Describe these studies that are related to your study. |
Response 58. Thank you for your comment. We updated the manuscript to include the necessary references. |
Comments 59. Page 10, paragraph ‘The study of depression, ...’ should be made more specific and not general. There is a lack of references to studies that prove that these measures are effective. |
Response 59. Thank you for your comment. We revised the paragraph and added relevant references. |
Comments 60. Add to the limitations already mentioned. Name the uncontrolled confounders. It then becomes clear that the study has major weaknesses. |
Response 60. Thank you for your comment. We revised the limitations section to specify uncontrolled confounders. We acknowledge that these factors represent significant weaknesses in our study and have discussed their potential impact on our findings. |
Comments 61. The conclusion only reports on results that are repeated. More focus should be placed on implications, relevance and the outlook for future research |
Response 61. Agree. Done. We added information: “Our study provides a critical foundation for future longitudinal research to further evaluate intervention effectiveness and address the underlying factors contributing to psychological distress in EMS personnel” |
Comments on the Quality of English Language Frequent spelling and grammar mistakes, inconsistent use of tenses, unclear translation of medical professions. A native speaker should be involved in the translation. |
Response: Thank you for pointing this out. We appreciate your suggestion and improve Language and Readability by using of journal’ English editing service. |

Round 2
Reviewer 3 Report
Comments and Suggestions for Authors
Thank you for dealing with my comments. I wish you all the best for your further research.
Comments on the Quality of English LanguageAs a non-native English speaker, it seems okay.